# Unravelling the amorphous structure and crystallization mechanism of GeTe phase change memory materials

Simon Wintersteller[1], Olesya Yarema[2], Dhananjeya Kumaar[1], Florian M. Schenk[1], Olga V. Safonova[3], Paula M. Abdala[4], Vanessa Wood[2] & Maksym Yarema[1] ✉

The reversible phase transitions in phase-change memory devices can switch on the order of nanoseconds, suggesting a close structural resemblance between the amorphous and crystalline phases. Despite this, the link between crystalline and amorphous tellurides is not fully understood nor quantified. Here we use in-situ high-temperature x-ray absorption spectroscopy (XAS) and theoretical calculations to quantify the amorphous structure of bulk and nanoscale GeTe. Based on XAS experiments, we develop a theoretical model of the amorphous GeTe structure, consisting of a disordered *fcc*-type Te sublattice and randomly arranged chains of Ge atoms in a tetrahedral coordination. Strikingly, our intuitive and scalable model provides an accurate description of the structural dynamics in phase-change memory materials, observed experimentally. Specifically, we present a detailed crystallization mechanism through the formation of an intermediate, partially stable 'ideal glass' state and demonstrate differences between bulk and nanoscale GeTe leading to size-dependent crystallization temperature.

Phase change memory (PCM) is a non-volatile memory technology which exploits the rapid and reversible switching capabilities of phase change materials to store information in their physical state. The crystallization and melting phase transitions between the high resistive amorphous state (logical 0) and low resistive crystalline state (logical 1) can be realized in a memory device through electrical or optical pulses inducing local heating. Fast switching speeds[1], ultra-low power consumption[2], and excellent stability[3] makes PCM a viable non-volatile alternative to present day's main memory (i.e., Dynamic RAM)[4], whilst outperforming storage class Silicon-based Flash memory with regards to access time, bandwidth, and cycling durability[5]. In addition to strong memory performance, the added value of PCM devices lies in its suitability for synaptic realizations in neuromorphic and in-memory computing[6,7]. Inspired by the energy-efficiency of the human brain,

where memory and processing are highly intertwined, arithmetic matrix-vector operations can be performed within the memory cell array by mapping matrix elements to the conductance values of the resistive memory device[8]. The key characteristic that enables synaptic realizations is the progressive crystallization of the PCM cell, where the conductivity of the phase change material is proportional to the degree of crystallinity[9]. Finally, PCM has shown excellent scaling into the sub-10 nm regime[10–12] in the form of nanoparticles[13,14] or ultrathin films[15], a key memory requirement to meet the exponentially increasing demands in data storage[16].

Despite excellent prospects of PCM technology including large-scale chip realization[5], in-memory computing[17] and neuromorphic applications[18–20], the fundamental understanding of the structure and switching mechanism in phase change materials remains largely

[1]Chemistry and Materials Design, Institute for Electronics, Department of Information Technology and Electrical Engineering, ETH Zürich, 8092 Zürich, Switzerland. [2]Materials and Device Engineering, Institute for Electronics, Department of Information Technology and Electrical Engineering, ETH Zürich, 8092 Zürich, Switzerland. [3]Paul Scherrer Institute, 5232 Villigen, Switzerland. [4]Laboratory of Energy Science and Engineering, Department of Mechanical and Process Engineering, ETH Zurich, 8092 Zürich, Switzerland. ✉e-mail: yaremam@ethz.ch

fragmented, even for commonly used chalcogenide glass compounds found within the Ge-Sb-Te phase diagram. This is particularly evident when studying the amorphous phase of PCM materials. While a variety of methods, such as x-ray absorption spectroscopy (XAS)[21–24], Raman spectroscopy[25,26] and ab-initio molecular dynamic simulations (AIMD)[27–30] have been used to study PCM materials, no universally accepted metrics exist yet to quantify the local structure and crystallization dynamics of the amorphous structure.

Up to date, most studies have focused on the practical question of PCM aging[31–35], i.e., the non-negligible increase of amorphous state resistivity in the PCM device during operation. The aging effect in PCM relates to structural relaxations in the amorphous phase as local structures transform into more energetically favorable coordination environments[31,32,36]. Such structure dynamics known as Johari-Goldstein relaxations (or β-relaxations), also occur in metallic glasses at temperatures below the glass transition[37–39]. For the case of GeTe PCM aging, Ge-Ge homopolar bonds are of high importance since they contribute a lone pair of electrons, thus increasing the charge carrier concentration in the amorphous structure[40–42]. Furthermore, Ge-Ge bonds stabilize tetrahedrally coordinated Ge atoms, which create in-gap and band tail defect states in GeTe. As a result, the observed decrease in conductivity (i.e., aging) is strongly associated with the breaking of Ge-Ge homopolar bonds[33]. The aged amorphous GeTe structure becomes progressively more alike the crystalline phase, resembling an 'ideal glass', with an increased fraction of 'ideal' three-fold coordinated Ge atoms and a narrower bond distribution[36]. These studies on PCM aging prelude many yet unanswered questions about the structure and dynamics of amorphous chalcogenides: Does crystallization and ultimate aging occur similarly? Are there any partially stable atomic arrangements (such as an 'ideal glass' state) upon crystallization? What are the nanoscale effects on crystallization and aging?

With these questions in mind, we turn to a prototypical phase change material, GeTe, which we can study across scales, comparing widely adopted sputtered GeTe microparticles with bulk-like properties and colloidally synthesized GeTe nanoparticles[13]. Through a series of isothermal high-temperature XAS measurements, we slow down the nanosecond crystallization process over the course of multiple hours, revealing a detailed crystallization mechanism. We argue that the crystallization phase transition starts with the structural pre-ordering of the Te sublattice and forms an 'ideal glass' intermediate state, from which the PCM material switches to its final crystalline phase. These experimental observations inspire a theoretical model of the amorphous GeTe structure, which can be explained as $sp^3$-hybridized organic-like Ge chains within a disordered *fcc*-type Te sublattice. While our simple model strongly matches the experimental data and previous literature on amorphous GeTe[33,41,43], it also provides a quantitative and intuitive understanding of the crystallization and aging processes. We conclude that GeTe phase transitions are, ideally, diffusionless for Te atoms, while Ge-Ge homopolar bonds cleave and form via Te-Ge-Te intermediate 'bridge' states. Furthermore, we derive the 'ideal glass' GeTe structure and reasons for higher crystallization temperatures in GeTe nanoscale materials.

## Results

We study two types of GeTe samples with distinctly different physical dimensions: GeTe microparticles, stemming from amorphous sputtered films, around 10–40 μm in size (Fig. 1a), and GeTe nanoparticles, a factor of 1000 smaller (Fig. 1b), with an average diameter of around 5 nm and narrow size distribution (Supplementary Fig. 1). We prepare GeTe nanoparticles through a colloidal hot-injection synthesis[13] and, for this study, replace the organic ligands with a monatomic ZnS shell, an effective low-cost capping material[44]. The ZnS thin shell (Supplementary Table 1) protects the nanoparticle core to minimize oxidation and coalescence processes, enabling us to reliably study nanoscale

crystallization effects at elevated temperatures for prolonged time periods. Both samples, which we will refer to as 'GeTe bulk' and 'GeTe nano', show good 1:1 stochiometric composition (Supplementary Table 1).

We begin our structural investigations with room temperature (RT) XAS measurements to study the local atomic structure of the amorphous and crystalline GeTe phases. The Fourier transform of the extended x-ray absorption fine structure (EXAFS) part of the XAS spectra reveals a radial distribution function of neighboring atoms around the central absorbing atom. Fitting the radial distribution function for both the Ge and Te K-edges enables us to assess the local atomic environment in GeTe, determining the coordination number and bond distances to the nearest neighbors (Supplementary Figs. 2–4). Figure 1c shows radial distributions (not corrected by phase) for the Ge K-edge, indicating a clear structure conversion from the amorphous to crystalline phase for both GeTe bulk and GeTe nano. In the amorphous GeTe phase, we observe a lack of long-range order with a maximum scattering path length of approx. 3 Å. In comparison, we note an expected long-range scattering intensity for the crystalline GeTe phase, obtained after heating amorphous samples to 400 °C, due to the increase in structural order. Critically, in both the amorphous and crystalline phase, the radial distribution functions of the bulk and nano GeTe are nearly identical (Fig. 1c). This brings us to the important conclusion that the structure of GeTe—whether amorphous or crystalline—remains the same across scales, even for the ultrasmall sub-10 nm dimensions.

### Structural units in crystalline and amorphous GeTe

From the RT EXAFS fits (Supplementary Tables 2–5), we can deduce the average structural unit for amorphous and crystalline GeTe (Fig. 1d)[21,32,41,45]. In the crystalline phase, GeTe forms a rhombohedral structure (Supplementary Fig. 5), where Ge and Te atoms are in the octahedral coordination, forming 3 shorter and 3 longer GeTe bonds ($d_{Ge-Te1} = 2.86$ Å; $d_{Ge-Te2} = 3.14$ Å)[46]. Our EXAFS fits agree with the known crystalline α-GeTe structure, showing similar bond distances for the bulk and nano GeTe (Fig. 1d). Every Ge atom has a total coordination of 6.3 for GeTe nano, while for GeTe bulk, the coordination number is 4.4, lower compared to a theoretical coordination of 6. (Supplementary Tables 2, 3). This difference comes from the higher estimation of mean square relative displacement (MSRD) values due to structural disorder and presence of Ge vacancies in the GeTe structure. Finally, the EXAFS fits in both cases clearly indicate a small fraction of homopolar Ge-Ge bonds (10-12 %) in crystalline GeTe, which is consistent with the literature[31,47].

In the amorphous GeTe phase, an average Ge atom forms an equal amount of homopolar Ge-Ge bonds and slightly longer Ge-Te bonds ($d_{Ge-Ge} = 2.46$–2.47 Å; $d_{Ge-Te} = 2.60$–2.62 Å) (Supplementary Tables 4, 5), resulting in an average structural unit resembling a distorted tetrahedron (Fig. 1d), which is supported by the literature concerning 4-fold coordinated Ge[21,45]. We also note larger MSRD values for the crystalline phase than for the amorphous phase, which we associate with lattice vibrations present in crystal structures strongly correlating to an increase in thermal disorder[48]. These lattice vibrations decrease with temperature and hence previous EXAFS studies of GeTe at T = 10 K have not shown this effect on the MSRD[49].

### High-temperature structural dynamics of GeTe

Having deduced the structural units in amorphous and crystalline GeTe, we now proceed to study their dynamics and conversion (i.e., crystallization) at high temperature. During the heating ramp XAS experiments (Fig. 2 and Supplementary Figs. 6, 7), crystallization can be clearly observed through the sudden increase in the coordination number of the Ge-Te bonds, occurring at around 200 °C for GeTe bulk and 250 °C for GeTe nanoparticles. Simultaneously, the Ge-Te bonds elongate as Ge atoms rearrange themselves into the center of Te

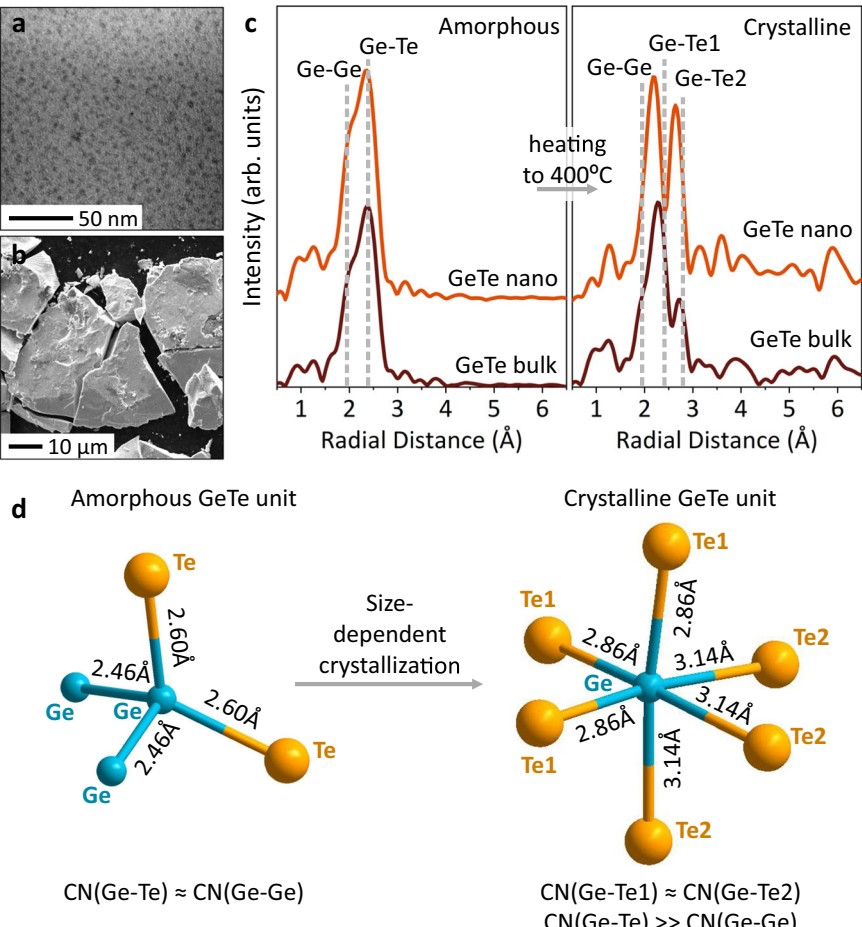

**Fig. 1 | GeTe phase-change material across scales. a** Scanning electron microscopy image of amorphous GeTe bulk microparticles and (**b**) transmission electron microscopy image of amorphous GeTe nanoparticles. **c** EXAFS radial distribution functions for GeTe bulk and GeTe nanoparticles, measured at room temperature before and after the heating ramp. **d** Average structural units for the amorphous and crystalline GeTe phases, as deduced from the XAS results in (**c**). CN denotes coordination number.

octahedra and split into two distinct Ge-Te bonds, matching the rhombohedral α-GeTe structure. Furthermore, we observe a decrease in the concentration of Ge-Ge homopolar bonds, indicating the breaking of Ge chains upon crystallization. GeTe nanoparticles crystallize in a similar manner despite slower changes in the path fitting parameters, which can be attributed to slower crystallization kinetics.

While the initial and final states of GeTe are merely the same for the bulk and nanoparticles (Fig. 1d), the crystallization process differs notably. Firstly, the crystallization temperature is higher for GeTe nanoparticles, also observed with in-situ x-ray diffraction (XRD) measurements (Supplementary Fig. 8). While we[13] and others[50,51] have reported size dependent GeTe crystallization temperatures, the structural origin has never been proposed for this phenomenon. Secondly, the structure dynamics near the crystallization temperatures differs between the bulk and the nano. For GeTe bulk, the structure remains relatively unchanged up until the crystallization temperature, where it switches abruptly to the stable crystalline phase (Fig. 2b–d). For GeTe nanoparticles, however, the structure dynamics is much less of a binary fashion. Long before the crystallization point, the coordination number of the second (longer) Ge-Te bond increases continuously with the temperature, while the amount of Ge-Ge bonds decreases (Fig. 2g). This indicates that a fraction of Ge atoms falls into more favorable coordination, which eventually stabilizes the GeTe structure at the nanoscale, delaying the crystallization to higher temperatures. We thereby discover the structural origin of non-negligible nanoscale effects in PCM materials, such as kinetics and mechanism of crystallization and aging.

## Understanding the amorphous GeTe structure

To assess the structural changes fully and quantitatively, we shall understand the structure of amorphous and crystalline GeTe. While crystalline GeTe can be presented as layers of 3-fold Te coordinated Ge atoms (i.e., half-octahedra), no consensus is reached for the amorphous GeTe yet[26,42,51]. Classical PCM materials, such as Ge-Sb-Te, are generally evaluated by measuring a variety of properties, such as the fraction of tetrahedrally coordinated Ge atoms, concentration of Ge-Ge homopolar bonds, length of Ge chains and formation of ABAB rings or density of voids[26,42,52]. In addition, representative approaches towards generating amorphous GeTe structures can be built using melt-quench AIMD simulations and neural network methods based on 1000 s of training sets[52,53]. Although powerful, many of these approaches are computationally heavy and fail to realize the realistic time scale which occurs during switching. Here, we seek to develop a simple yet scalable model for the amorphous GeTe, which supports our experimental findings as well as prior structural knowledge.

To start off, we note that the amorphous GeTe unit (Fig. 1d) is highly reminiscent to a basic −$CH_2$− unit in the alkane homologous organic series. The central Ge (as much as C in organic molecules) has a coordination number of 4, being tetrahedrally coordinated and $sp^3$ hybridized. Figure 3a elaborates on this obvious similarity. Except for methane, C atoms have two types of neighbors and the average ratio

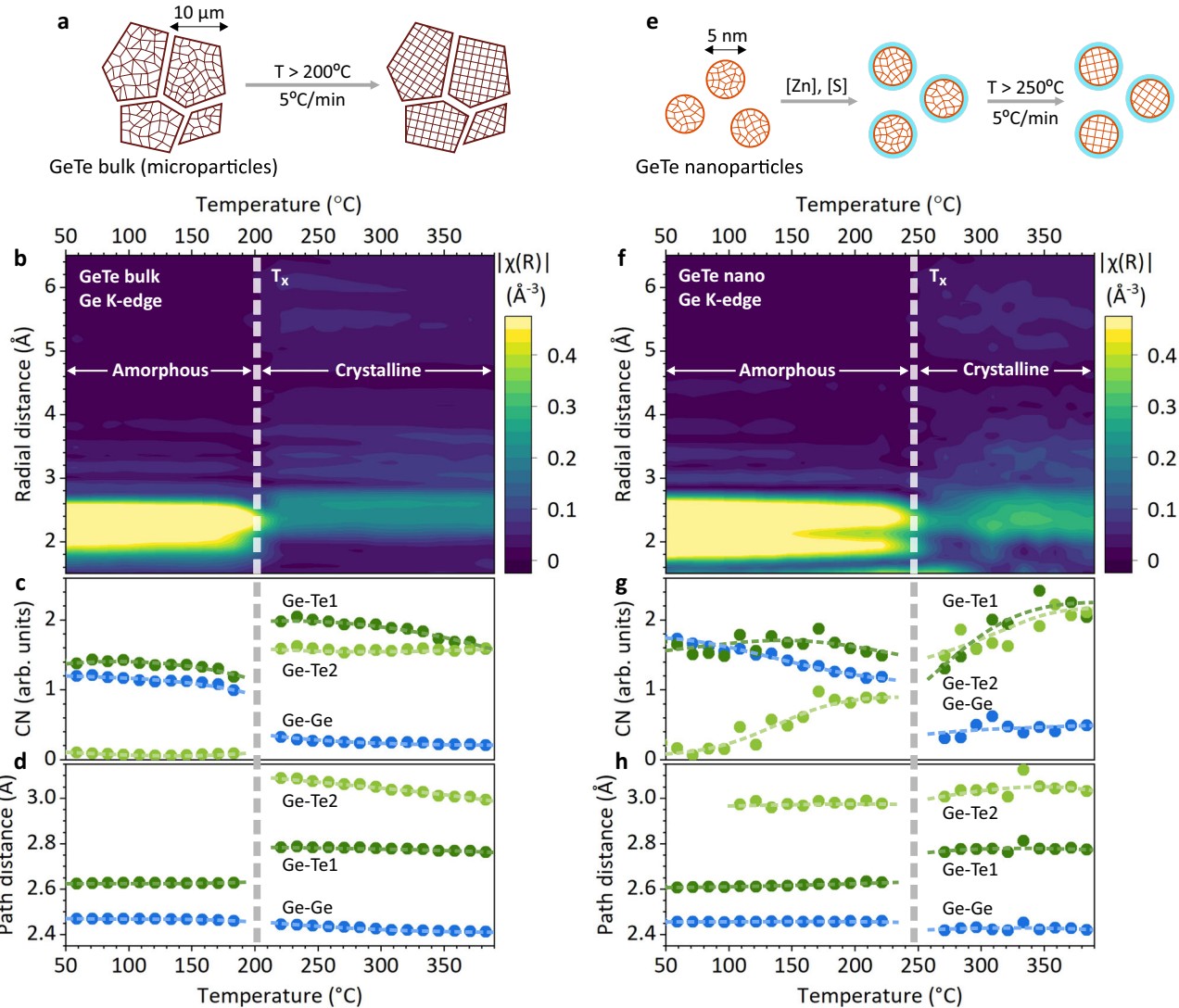

**Fig. 2 | High-temperature structural dynamics of GeTe. a** Schematics of 5 °C/min in-situ heating ramp measurements for GeTe bulk and EXAFS fitting results, including radial distribution functions (**b**), global coordination numbers (**c**), and path distances (**d**) as a function of temperature. **e–h** The same data for GeTe nanoparticles. CN denotes coordination number.

between C-H and C-C coordination numbers decreases as the chain becomes longer. For example, in butane an average C atom is neighbored with 2.5 atoms of H and 1.5 atoms of C, while for an infinitely long polymer chain, the C-H and C-C coordination numbers are both equal to 2 (Fig. 3a). This closely matches our amorphous GeTe unit (Fig. 1d) as derived from XAS measurements, where every Ge atom is bonded to 2 other Ge and 2 Te atoms (Fig. 2, Supplementary Tables 4, 5). Furthermore, Ge atoms are known to form complex chains within the amorphous phase[42], further supporting a comparison between Ge chains and organic-type polygermanes[54–56].

Concluding that Ge chains are the building blocks in the amorphous GeTe structure, we proceed to study the inverse problem of crystallization, i.e., the easiest way Ge-Ge 'polymer-like' chains can be formed from the crystalline GeTe phase. We note that H atoms arrange as side-sharing distorted octahedra in organic alkanes, such as butane (Supplementary Fig. 9). This atomic arrangement is also characteristic for the Te sublattice in crystalline GeTe, except that for organics each $H_6$ octahedron hosts 2 C atoms, while $Te_6$ octahedra host only 1 Ge atom. Stemming from this analogy, we hypothesize that Ge chains can be formed in a diffusionless fashion, preserving the original *fcc*-type Te sublattice. Figure 3b shows schematically a

(100) plane of crystalline GeTe to illustrate the formation of Ge chains via tunneling of Ge atoms and a small distortion of the Te sublattice. Figure 3c presents atomic displacement histograms for the formation of a $Ge_4$ chain (Fig. 3b). Overall, 14 atoms change their positions with respect to the crystalline GeTe structure (Supplementary Fig. 10). Te atoms experience only relatively small shifts on the order of 1.0–1.5 Å. Smaller Ge atoms, on the other hand, must move approx. 2 Å within the octahedra to accommodate the incoming Ge atoms, tunneling approx. 3.5 Å from the neighboring octahedra. This enables us to create a model that is highly scalable (Fig. 3d) and can be seen as a representation of the amorphization process. In the amorphous structure, Ge atoms cluster into chains with various lengths and geometry, whilst at the same time, the Te sublattice maintains a long-range *fcc*-type arrangement requiring only a small atomic displacement upon amorphization (Fig. 3e). Our model compliments the long-standing hypothesis about structural closeness between amorphous and crystalline GeTe[57]. We also show that the amorphization and crystallization processes can theoretically occur without the formation of unfavorable energy-expensive antisite defects. Combined, our simple model explains why the phase transitions of PCM devices are rapid and highly durable[58–60].

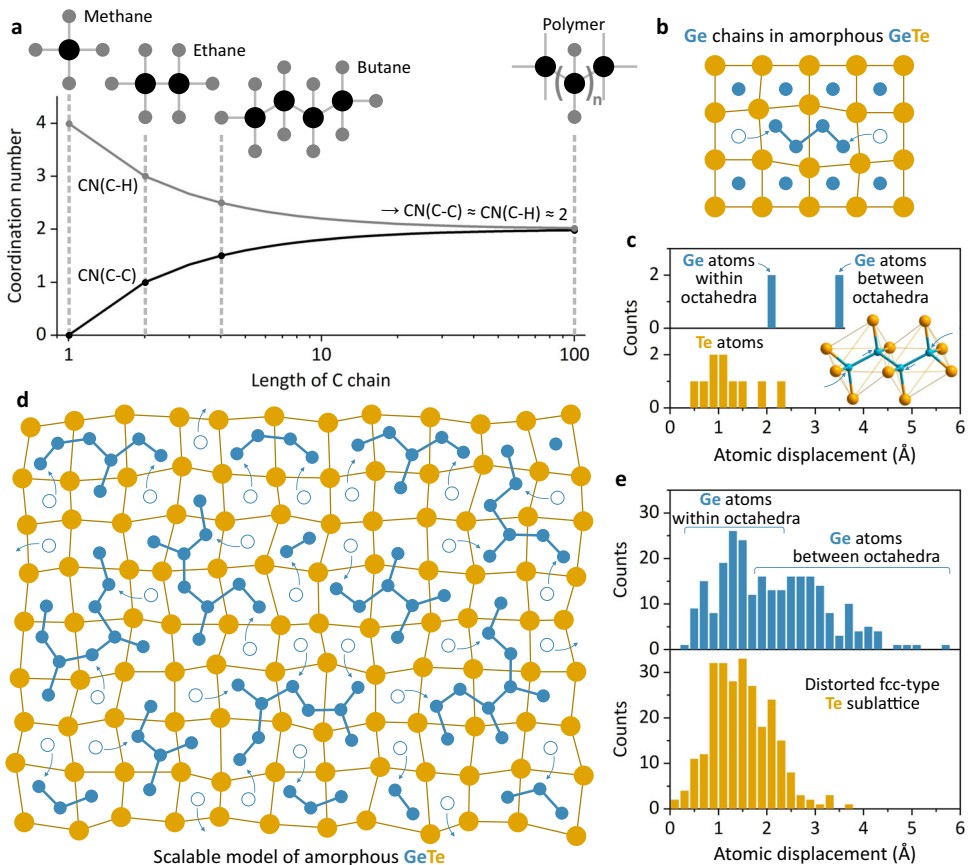

**Fig. 3 | Modelling the structure of amorphous GeTe. a** Coordination number analysis for the saturated hydrocarbons with different C chain length and (**b**) formation of a Ge₄ chain within the crystalline GeTe structure, shown schematically for the (100) cubic GeTe plane. **c** Atomic displacement histograms of the Ge₄ chain, shown in (**b**) and inset (**c**). **d** 2D depiction of an upscaled GeTe amorphous model to illustrate the amorphization process and (**e**) associated atomic displacement histograms for the DFT-relaxed 512-atom model. C atoms are in black, H in grey, Ge in blue, and Te in orange.

In the next step, we present a methodology to generate realistic GeTe amorphous structures in order to study the local coordination and relate it to our experimental findings and previous literature. We start with Te atoms in their positions as for the crystalline β-GeTe phase and populate half of Te₆ octahedra with two Ge atoms. This approach creates a random network of ethane-like Ge₂Te₆ structural units, except that each Te atom belongs to two Ge₂Te₆ structural units, rendering a GeTe stoichiometry of 1:1 (Supplementary Fig. 11). This starting structure is then relaxed using geometry-optimized DFT calculations, converging to a final amorphous state (Supplementary Fig. 12). Figure 4a shows the result for the 512-atom amorphous GeTe, highlighting the random arrangement of Ge chains with various length and branch networks. To validate our amorphous model, we calculate its pair distribution function (PDF) and compare it to experimental PDF analysis of x-ray total scattering data of the amorphous and crystalline GeTe bulk microparticles (Fig. 1b). Figure 4b clearly demonstrates the amorphousness of the simulated GeTe structure, lacking any degree of long-range order above 7 Å. In fact, our amorphous GeTe model has an even higher extent of disorder in comparison to the measured sample, leading to smaller intensities of e.g., the Te-Te peak around 4.1 Å or the Ge-Te peak around 6.4 Å. Overall, we observe a strong correlation between experimental PDF of amorphous GeTe and independently simulated PDF data, while a small 0.1 Å shift in the initial peak position suggests a larger amount of Ge-Ge bonds i.e. longer Ge chains, present in the amorphous GeTe bulk sample.

To quantify the generated amorphous GeTe structure, we determine bond distance cutoff values from the simulated PDF curves (Fig. 4b). We assume the minimal value following the main peak in

bond-specific PDFs as a cutoff bond distance, namely 2.8 Å for the Ge-Ge and 3.0 Å for the Ge-Te bonds. For the Te-Te bonds, a cutoff distance of 4.7 Å relates to the second coordination sphere (i.e., distorted *fcc*-type Te sublattice), while smaller bond cutoff distance of 3.1 Å has to be chosen to quantify homopolar Te-Te bonds. We then study the bond order distribution (BOD), and bond angle distribution (BAD) for Ge and Te, revealing the main building blocks of amorphous GeTe. The BOD analysis shows that Ge atoms are mostly 3-fold and 4-fold coordinated, while Te atoms have either 2 or 3 neighbors (Fig. 4c, d). We can further deconstruct the coordination number histograms and thus identify the neighbors for each configuration and its occurrence in amorphous GeTe (right panels of Fig. 4c, d). For 4-fold coordinated Ge, the most common structural unit is a tetrahedron with one Ge and 3 Te neighbors, followed by tetrahedra with 2 and 3 Ge atoms in the first coordination sphere (Fig. 4c). In view of organic-type Ge ordering, these units correspond to the end, the middle, and the branching atom in the Ge chain, respectively. Ge atoms with 3 neighbors are either fully coordinated with Te or have 1 Ge and 2 Te atoms in the local environment. While the former unit is characteristic for the crystalline GeTe state, the latter can be seen similarly to the unsaturated alkene hydrocarbon (Fig. 4c). Te atoms are more homogeneously distributed with only around 17% containing Te-Te homopolar bonds (Fig. 4d). A large fraction of Te atoms forms Ge-Te-Ge bridge units (2-fold coordination) interconnecting neighboring Ge chains, with the remaining Te atoms being 3-fold coordinated and surrounded by Ge atoms, similar to the crystalline GeTe state (Fig. 4d).

The BAD analysis (Fig. 4e) gives further insight into the local structure of amorphous GeTe. We observe that homopolar Ge-Ge-Ge

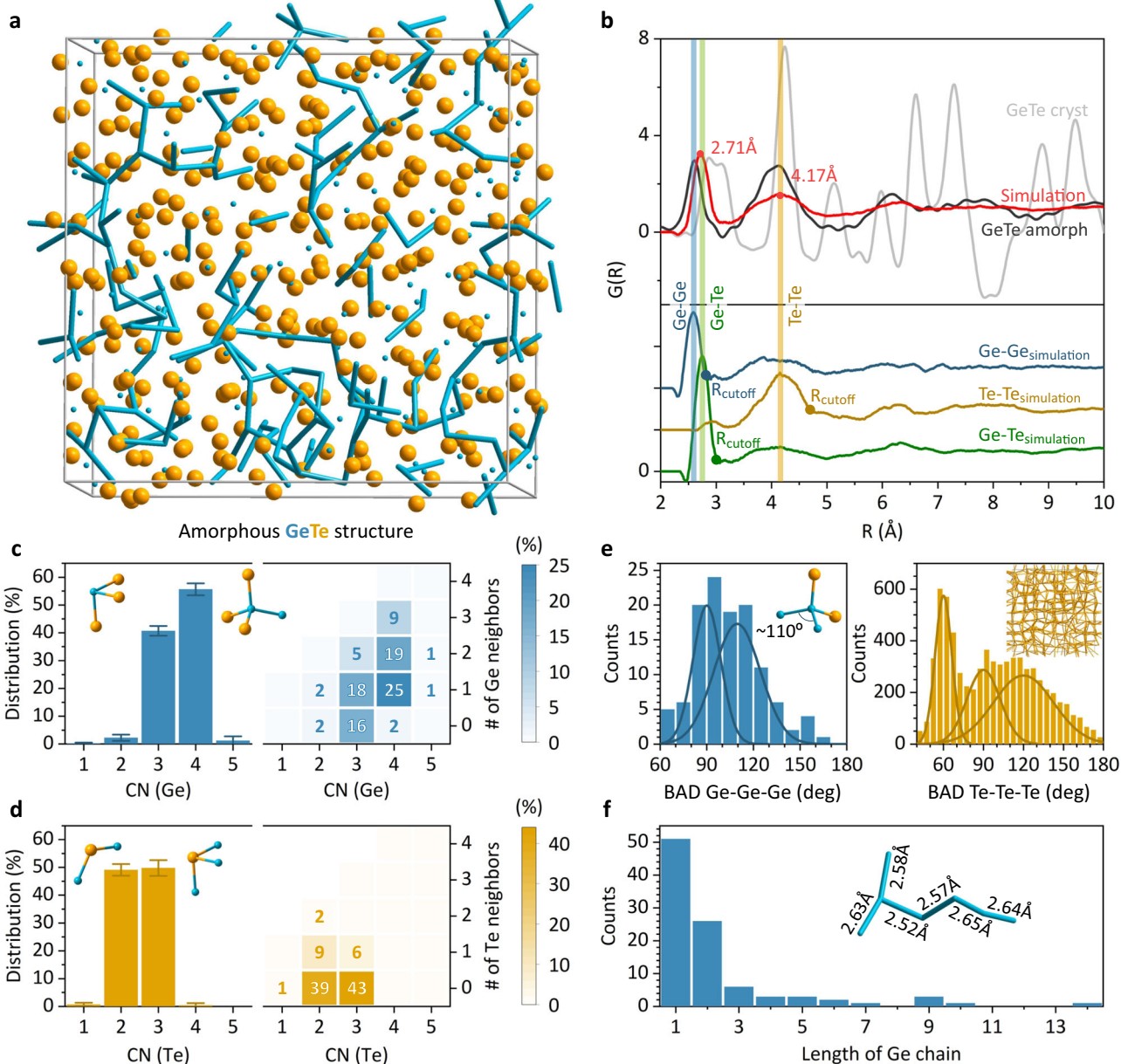

**Fig. 4 | Quantifying the scalable amorphous GeTe model. a** 3D representation of the 512-atom amorphous GeTe model after DFT structural relaxation, highlighting the complex network of intertwined Ge chains. **b** Pair distribution functions of the simulated GeTe structure and experimental data for the amorphous and crystalline GeTe bulk microcrystals. **c**, **d** Bond order distribution analysis of Ge and Te local coordination environment. Error bars represent standard deviation across four 512-atom amorphous structures. Numbers on the heatmaps denote a fraction of structural units (in %) with defined coordination number and neighboring atoms. Insets are the most common structural units in the amorphous GeTe structure. CN denotes coordination number. **e** Bond angle distributions (BAD) for the Ge and Te sublattices (i.e., Ge-Ge-Ge and Te-Te-Te bond angles) with structural motifs of the $sp^3$ hybridized Ge atom and distorted *fcc*-type Te sublattice, shown as insets. **f** Length distribution of Ge chains in the amorphous GeTe structure and an example of a $Ge_6$ chain shown as inset. Ge atoms are in blue and Te in orange.

bonds have a broad distribution of bond angles, which can be described with 2 peaks centering at 90° and 110°. This compliments the BOD analysis, supporting that Ge atoms contribute to the formation of short and strong $sp^3$ hybridized homopolar Ge-Ge bonds. This is in stark contrast to Te-Te-Te bonds, forming much sharper BAD around 60° bond angles, scaffolding the amorphous GeTe structure with underlying disordered *fcc*-type Te sub-lattice (Fig. 4e).

Finally, we study the distribution of Ge chains in our structure (Fig. 4f). Most of Ge chains are relatively short with maximum chain lengths of 2–6 Ge atoms. A few chains, however, are significantly longer, containing up to 14 interconnected Ge atoms in the longest

path as well as highly branched morphologies (Supplementary Fig. 13). Importantly, our quantifications of amorphous GeTe model are in line with the literature, in which BAD, BOD and PDF analysis have been typically performed after ab initio molecular dynamic simulations (Supplementary Figs. 14, 15)[28,29]. Extracted electronic and vibrational properties (Supplementary Figs. 16–18) point to the large extent of amorphousness in our GeTe structures. Hence, we have provided an intuitive, bottom-up approach to model amorphous PCM materials. Stemming from simple principles of organic and inorganic chemistry, our modelling method is highly scalable, and it requires significantly less computational power than AIMD approaches. Most importantly, we show in the next sections that the model can capture the structural

dynamics phenomena, such as crystallization mechanism, aging process, and nanoscale effects in PCM materials.

## High-temperature structure dynamics of amorphous GeTe

We return to in-situ XAS measurements, taken for both GeTe bulk microcrystals and GeTe nanocrystals as they are annealed at

temperatures below the crystallization points (Figs. 5a, b). We choose relatively high annealing temperatures (170 °C for GeTe bulk and 200 °C for GeTe nano) to optimize the XAS measurement time, but also to keep significant temperature offsets (i.e., 30 °C below the crystallization point for GeTe bulk and 50 °C offset for GeTe nano) to slow down the structure dynamics to last between 1 and 2 h. As a result,

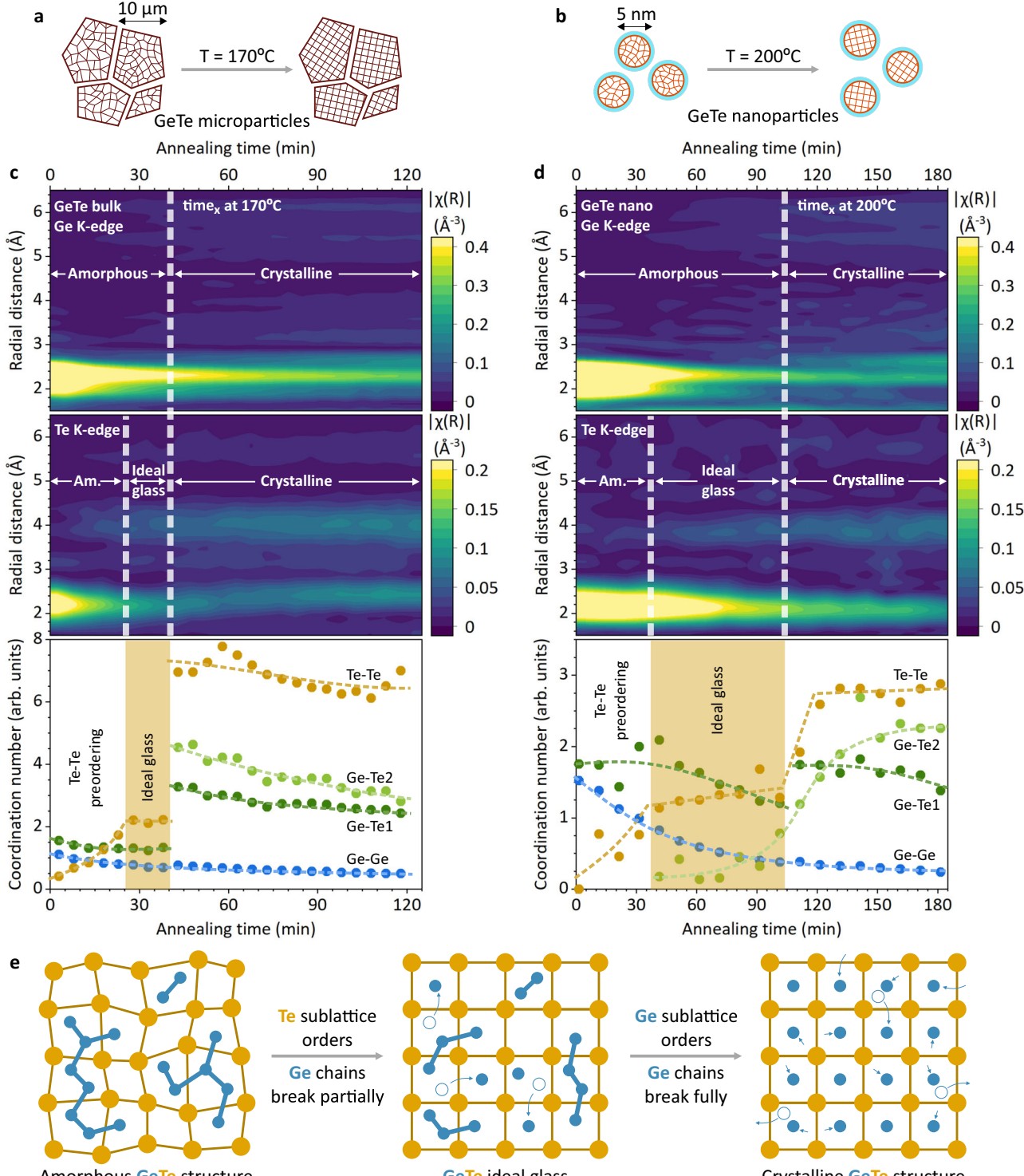

**Fig. 5 | Structural dynamics and crystallization mechanism of GeTe.**
**a**, **b** Schematics of annealing experiments for GeTe bulk and GeTe nanocrystals and (**c**, **d**) in-situ XAS measurements at the Ge K-edge and Te K-edge and EXAFS fitting results. Markups denote the stages of GeTe crystallization process: amorphous, partially crystalline 'ideal glass', and crystalline states. **e** Proposed crystallization mechanism in GeTe, based on macroscopic atomic model of the amorphous structure and XAS results in (**c**, **d**).

we capture the structure evolution in fine detail and distinguish between Ge and Te local bonding dynamics (Fig. 5c, d and Supplementary Figs. 19, 20).

For bulk GeTe, crystallization commences via initial preordering of Te atoms, visible through the fast and steady increase in the coordination number of the Te-Te bond (Fig. 5c and Supplementary Fig. 21). Crucially, this change in the Te sublattice is notably more significant than the structural changes on the Ge edge (i.e., modest decrease of the Ge-Ge bonds). After initial ordering, the Te sublattice forms an intermediate state, which remains visibly stable at high annealing temperatures until the amount of Ge-Ge bonds reaches critically low concentrations. Afterwards, the structure undergoes a rapid binary switch to the crystalline phase, indicated by the sudden increase in the coordination number of Ge-Te and Te-Te bonds around 40 min (Fig. 5c). At the nanoscale, GeTe experiences all the same stages upon crystallization: preordering of Te sublattice, followed by stable intermediate state and an abrupt switch to the crystalline state (Fig. 5d).

All in all, it suggests the universal crystallization mechanism for GeTe across scales. We employ our GeTe amorphous model to explain the crystallization process (Fig. 5e). Initial preordering of Te sublattice is naturally fast, because Te atoms maintain long-range *fcc*-type ordering in amorphous GeTe (Fig. 5e). Upon Te ordering, $Te_6$ octahedra with 2 Ge atoms become less distorted and thereby smaller in volume, leading to the accelerated breaking of Ge chains. Finally, as more Ge atoms occupy their crystalline positions, crystallization centers (e.g., $Ge_4Te_4$ cuboids) appear[61,62], leading to a rapid phase transition. Therefore, our viewpoint on Te-Te preordering provides a missing puzzle, which is complementary to the crystallization mechanisms, described via ordering of Ge and Te first coordination sphere (e.g., reorientation of ABAB rings).

We note that the stable intermediate state requires around a third of Te atoms to order with respect to the final crystalline state (Fig. 5c). We argue that this structure can be regarded as ideal GeTe glass, in which the Te sublattice becomes partially crystalline as schematized for the (100) rock salt plane in Fig. 5e. More precisely, however, we suppose that Te atoms define a network of structural units in the ideal GeTe glass, in which 4-out-of-12 Te neighbors arrange in a crystalline fashion (Supplementary Fig. 22). Simultaneously, Ge atoms form short chains within the semirigid Te network, such as the shortest ethane-like $Ge_2Te_6$ unit, observed previously in $Cr_2Ge_2Te_6$ PCM material[63].

Taking a closer look at the structural dynamics of GeTe nanoparticles, we note a slow increase in the coordination number of elongated Ge-Te bonds prior to crystallization (Fig. 5d), which we also observe during the XAS ramp measurements (Fig. 2g). Absent in the bulk material (Figs. 2c, 5c), these new local environments in nanodimensional GeTe appear to stabilize the intermediate ideal glass GeTe state (Fig. 5d). This effectively delays the final transition to the crystalline GeTe structure, which explains the higher crystallization temperatures observed in GeTe nanoparticles[13]. The more stable ideal glass state, however, points to slower crystallization kinetics, yet better data retention and aging properties for GeTe nanoparticles. To visualize this new local bonding, we employ our GeTe model (Fig. 4a), in which we observe many Ge atoms with coordination number of 5, if the cutoff value for Ge-Te bond is increased by 10% (Fig. 6). We argue that these local bonds can be 'locked' for GeTe nanoparticles, due to slower atomic mobility at the nanoscale[15]. Consequently, the crystallization phase transition for GeTe nanoparticles must include intermediate Ge states with a coordination number of 5, which is in contrast to GeTe bulk, where tetrahedrally-coordinated Ge switches directly to the octahedral environment (Fig. 6).

## Discussion

A previous study on β-relaxations in PCM materials established a direct correlation between weaker β-relaxations and slower crystallization kinetics[39]. Weaker β-relaxations correlate to reduced atomic mobility,

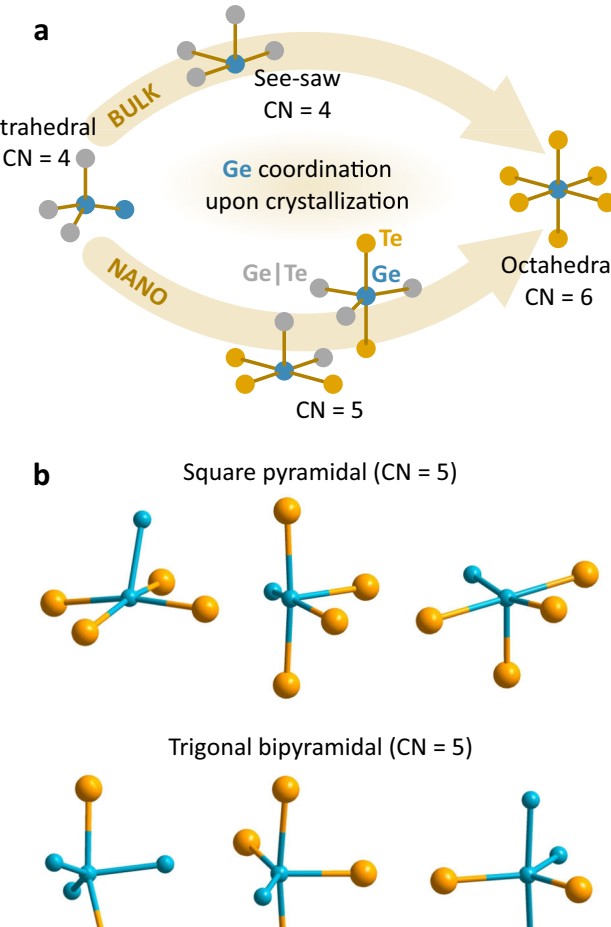

**Fig. 6 | Nanoscale effects for the amorphous GeTe. a** Comparison of local coordination environment for GeTe bulk and nanoparticles during the crystallization switching and (**b**) examples of stable local Ge bonding with coordination number of 5, occurring only for nanodimensional GeTe materials. Ge atoms are in blue and Te in orange. CN denotes coordination number.

which is highly desirable for PCM devices, resulting in a higher stability of the amorphous phase and slower aging processes[39]. The latter has already been shown for nanoscale PCM materials, comparing both 3 nm GeTe and Sb ultrathin films to the bulk[15,64]. Verified through simulations[38], decreased atomic mobility for the ultrasmall Sb has been associated with higher lattice stress induced through spatial confinement and $(ZnS)_8(SiO_2)_2$ capping. We observe the same trends for the ZnS-coated GeTe nanoparticles whose amorphous state appears to be more stable at the same conditions of elevated temperatures than in the case of its bulk counterpart (Fig. 2). This can be explained using the previous deductions: the ZnS shell increases surface strain which exerts higher stresses within the nanoparticle leading to lower atomic mobility and hence stronger covalent bonding and Ge chains. This explains why highly confined GeTe nanoparticles, like ultrathin films, exhibit suppressed aging[64], providing a roadmap for improved data retention and multibit storage in sub-10 nm PCM technology.

Furthermore, we also observe a slower transition from the relaxed 'ideal GeTe glass' state to the final crystalline state in GeTe nanoparticles in comparison to bulk GeTe (Fig. 5). This can be explained from the lower EXAFS-evaluated MSRD values, a measure of both static and thermal disorder, for GeTe nanoparticles compared to bulk (Supplementary Tables 2–5). Since nanoparticles inherently have higher static disorder, primarily to surface defects, a lower thermal disorder at the nanoscale must be due to stronger bond covalency. The

lower thermal disorder also correlates with reduced atomic mobility[65], which links back to our conclusions above. Finally, the reduction of atomic mobility in glassy chalcogenides leads to an increase in the glass transition temperature[65], providing an explanation for higher crystallization temperatures in nanoparticles[13].

We conclude that weak β-relaxations are highly desirable for embedded memory applications, especially if long-term data retention and reliable analog-type multibit data storage are of prime importance. High glass transition and crystallization temperatures comes with the benefits of data non-volatility at high operating temperatures, extending the target applications of PCM technology to the automotive industry and space exploration. Although the weaker β-relaxations are associated with slower crystallization kinetics and thus may initially appear as a disadvantage in confined nanoparticles, when realized in memory devices, this disadvantage is offset by improved thermoelectric effects[66,67] and size-dependent volumetric switching parameters, linked with ultrasmall sub-10 nm dimensions of active PCM memory bit.

In summary, this work studies and compares GeTe bulk microparticles with GeTe nanoparticles, revealing the structure and high-temperature dynamics via in-situ XAS experiments and theoretical modelling. We show that GeTe bulk and nanoparticles have the same amorphous phase, consisting of a disordered *fcc*-type Te sublattice, in which Te atoms link intertwined Ge chains, whose morphology is reminiscent to organic-like polygermanes. We perform annealing XAS measurements at temperatures below the crystallization point of GeTe, allowing us to unravel a 'macroscopic' crystallization switching mechanism with fine details. We demonstrate that GeTe starts crystallizing through the initial preordering of the Te *fcc*-type sublattice, a diffusionless process requiring minimal activation energy. Then, GeTe forms an intermediate ideal glass state, which remains intact until enough thermal energy is acquired to break the strong Ge chains, allowing Ge atoms to tunnel into vacant octahedral centers forming the final crystalline phase. We highlight a pivotal role of Te sublattice in the crystallization mechanism and argue that the structural relaxations and preordering upon crystallization are the same β-relaxations observed during aging. Finally, we unravel the nanoscale effects on crystallization and aging of GeTe. Reduced atomic mobility, due to nanoparticle confinement and surface stresses, weakens β-relaxations explaining a more stable amorphous structure, favorable local bonding, and higher crystallization temperatures in GeTe nanoparticles. Understanding the structural dynamics during crystallization will aid the development of improved material compositions as well as superior design of PCM devices, improving the control and switching mechanism, enabling the realization of innovative memory applications, and reaching ultrasmall sub-10 nm memory cells via rational design.

## Methods

### Materials
GeI$_2$ (99.99%) was purchased from ABCR, tri-*n*-octylphosphine (TOP, 97%) and Te (broken ingots, 99.999%) from STREM, oleic acid (90%), chloroform (99%), ethanol (99.8%), diethylzinc solution (1.0 M in hexanes), sulfur (99.98%) from Sigma-Aldrich, lithium bis(trimethylsilyl) amide (LiN(SiMe$_3$)$_2$, 95%) from Acros Organics. Oleic acid was dried at 100 C for 1 hour from water residues and all other chemicals were of anhydrous grade and were used as received. Anhydrous boron nitride (BN) from Alfa Aesar was dried under vacuum in a squalene oil bath at 250 °C for 24 h.

### Synthesis of GeTe/ZnS core/shell nanoparticles
GeTe nanoparticles (5 nm in diameter) were synthesized using the colloidal synthesis method described previously[13]. After the synthesis, oleic acid ligands were removed and replaced with a protective ZnS shell through a cation-exchange reaction. Specifically, 30 mg of GeTe

nanoparticles in hexane were mixed with 10 ml of tri-*n*-octylphosphine (TOP) and 2 ml of 1.0 M sulfur solution in TOP (i.e., TOP:S). The mixture was stirred under vacuum for 30 min to evaporate the hexane and other volatile residuals. Afterwards, a mixture of 0.5 ml 1.0 M diethylzinc solution in hexane, 0.5 ml of TOP:S, and 5 ml of TOP was added dropwise at 80 °C. The mixture was retained at 80 °C for an additional 10 min, cooled down to room temperature, and purified using hexane/ethanol solvent/non-solvent washing cycles. After the first washing step, a small amount of oleic acid was added to compensate detached surface surfactants.

### Amorphous GeTe microparticles
GeTe thin films were produced by dc magnetron sputtering) with stoichiometric targets of 99.99% purity. After sputtering, the as-deposited films (1 - 10 μm) were exfoliated from Si substrates and milled into powders to perform subsequent analysis.

### Structural characterization
All sample preparation was carried out under inert N$_2$ atmosphere to prevent any sample oxidation. For X-ray absorption spectroscopy (XAS) and high temperature X-ray diffraction (HT-XRD) measurements, samples were mixed with anhydrous boron nitride using a 20−40 wt.% concentration, before being loaded into 1.5 mm quartz capillaries and sealed air-tight using epoxy resin. For pair distribution function (PDF) measurements, sputtered amorphous GeTe was filled into 0.3 mm quartz capillaries under N$_2$ atmosphere and sealed air-tight using epoxy resin.

Transmission electron microscopy images were taken with a JEM-1400 Plus JEOL instrument operated at 300 kV. Scanning electron microscopy images and. energy dispersive x-ray (EDX) spectroscopy was performed on an FEI Quanta 200 F SEM microscope equipped with an Octane Super EDX detector and operated at 30 kV. High-temperature X-ray diffraction measurements were performed on a Rigaku SmartLab 9 kW system, equipped with a rotating Cu anode and 2D solid state detector (HyPix-3000 SL) and a high-temperature stage (HT1100 Anton Paar). XRD measurements were performed using a parallel beam geometry with a step size of 0.01° and scanning speed of 5°/min. Sample filled capillaries were placed flat on top of a ceramic heating plate and measurements were performed under a constant heating ramp of 5 °C/min up to 400 °C with a temperature precision of ±1 °C. Atomic pair distribution function (PDF) analysis was performed using X-ray total scattering data acquired on a laboratory goniometer-based X-ray scattering instrument setup (Empyrean, Malvern Panalytical). The transmission setup utilized a sealed X-ray tube with a silver target (λ = 0.56 Å for Ag Kα), and a hybrid pixel detector GaliPIX with a CdTe sensor[68]. Data from an empty capillary was subtracted from the scattering signal of the samples. The total acquisition time was 24 h (12 scans of 2 h per scan were averaged) with a step size of 0.07°. The respective pair distribution functions, G(R), were obtained using the High Score software by using the Fourier transform of the normalized X-ray scattering function (F(Q), Supplementary Fig. 23), using $Q_{min} = 1$ Å$^{-1}$ and $Q_{max} = 20$ Å$^{-1}$ ($Q = 4\pi \cdot \sin\theta/\lambda$).

X-ray absorption spectroscopy (XAS) experiments were performed at the SuperXAS beamline of the Swiss Light Source at the Paul Scherrer Institute in Villigen, Switzerland. The Swiss Light Source operates in top-up mode at 400 mA and 2.4 GeV. XAS spectra were collected at the Ge K-edge between 10,800 and 12,600 eV and at the Te K-edge between 31,400 and 32,500 eV using a Si(111) and Si(311) channel-cut QuickXAS monochromator oscillating at 1 Hz frequency[69]. The L3 edge of a Platinum foil (11563.7 eV) and the K-edge of a TeO$_2$ pellet (31818 eV) were used to calibrate the Ge K-edge and Te K-edge, respectively. For temperature measurements, an in-house capillary reactor setup was used where the sample capillary was placed in between two infrared heaters. For accurate temperature control, a

thermocouple was placed in a BN filled capillary and measured in parallel to the sample (Supplementary Fig. 24). An in-house python based "ProQEXAFS" v.2.43 software[70] was used for data importing and energy calibration. For ramp measurements, XAS spectra were averaged every 2.5 min and for annealing measurements a 5 and 10 min moving average with 10 and 20 min averaging windows was used for the microparticles and nanoparticles respectively.

### Analysis of XAS data
The extended X-ray absorption fine structure (EXAFS) part of XAS spectrum was analyzed using the Demeter software package[71]. The crystalline α-GeTe rhombohedral structure[46] was used as the theoretical model to perform the EXAFS fitting for the two Ge-Te and two Te-Te shells. In addition, a tetrahedrally coordinated Ge atom with a single Ge neighbor at a bond distance of 2.605 Å, obtained from our amorphous structure, was used to model the Ge-Ge shell. For ramp measurements, only the Ge edge was measured and hence only the first nearest neighbors were fitted, i.e. Ge-Ge, and the two Ge-Te shells. For room temperature and annealing measurements, both Ge and Te K-edges were measured, permitting the additional fitting of the Te second coordination sphere (i.e., the two Te-Te shells). In addition, measuring at two edges permitted the coupling of fitting parameters such as the bond coordination number and path distance. The EXAFS part of the Ge K-edge XAS spectra was Fourier transformed using a k-weighting of 2, k-range of 3-12 Å$^{-1}$ 2dk and Kaiser-Bessel window. An R-range of 1.7–3.5 Å was used to fit the Ge-Ge, Ge-Te1, and Ge-Te2 bonds. For the Te K-edge, the Fourier transform was performed using a k-weighting of 2, k-range of 3-12 Å$^{-1}$ 2dk and Kaiser-Bessel window. An R-range of 1.7–5.0 Å was used to fit the Ge-Te1, Ge-Te2, Te-Te1, and Te-Te2 bonds.

A single structural fitting model was used to evaluate the vastly different coordination environments of the amorphous, crystalline, and intermediate states. The amplitude reduction factors ($S_0^2$), 0.792 and 0.905 for the Ge and Te edge respectively, were determined from Ge bulk and Te bulk reference measurements. Furthermore, a global edge-dependent $\Delta E_0$ was fitted for every evaluated spectrum. A variable parameter was assigned to the coordination number of the different bonds whilst restrictions on the mean square relative displacement (MSRD), to account for thermal and static disorder, were placed. Initial MSRD values for the Ge-Te1, Ge-Te2 Te-Te1, and Te-Te2 bonds were obtained from EXAFS measurements on crystalline GeTe measured at 10 K[49]. In order to account for the increase in disorder at higher temperatures (i.e., at room temperature or during ramp and annealing measurements), the initial MSRD estimates ($3.1 \cdot 10^{-3}$, $4.9 \cdot 10^{-3}$, $4.5 \cdot 10^{-3}$, and $5.5 \cdot 10^{-3}$ Å$^2$ for Ge-Te1, Ge-Te2, Te-Te1, and Te-Te2 respectively) were multiplied by a variable, edge dependent scaling parameter (Ge/Te σ$^2$ scaling factor). Furthermore, where relevant, the coordination number and path distances were coupled to both Ge and K edges. To reduce an additional fitting parameter, a single variable parameter was assigned to the coordination number of both Te second nearest neighbors (i.e., Te-Te1 and Te-Te2) and labelled Te-Te. No significant increase in bond disorder was observed at higher temperatures for the short homopolar Ge-Ge bonds and hence this path was always fitted with a fixed MSRD of $3.7 \cdot 10^{-3}$ Å$^2$ and variable coordination number. In total, for every fit performed on the Ge and Te edge there were 13 variable parameters: $\Delta E_0$ Ge, $\Delta E_0$ Te, Ge σ$^2$, Te σ$^2$, CN Ge-Ge, CN Ge-Te1, CN Ge-Te2, CN Te-Te, R Ge-Ge, R Ge-Te1, R Ge-Te2, R Te-Te1, Te-Te2.

### Theoretical modelling
512-atom amorphous supercell structure consisting of a network of 4 × 4 × 4 Te octahedra was generated, representing a *fcc* Te sublattice in crystalline GeTe phase with the 6.3311 Å lattice constant (a total of 256 Te atoms). Half of the Te$_6$ octahedra were then populated with two Ge atoms, spaced 2.43 Å apart and with a random Ge-Ge bond

orientation (a total of 256 Ge atoms). The stoichiometry of a GeTe model was therefore strictly 1:1 Ge:Te. The final structure has an atomic density of 0.031524 atoms·Å$^{-3}$ corresponding to the same atomic density determined in other amorphous ab-initio molecular dynamic simulations[52]. We generated 5 different starting GeTe structures to account for statistical variations in the initial arrangement of vacant and double populated Te octahedra.

### DFT structure relaxation and calculations
The generated GeTe structures were then relaxed using density functional theory (DFT) calculations in order to obtain the final amorphous structures. DFT calculations were performed within the open-source CP2K program[72]. Calculations were done using a dual basis of localized Gaussian and plane waves with a 300RY plane wave cutoff. Triple-Zeta-Valence-Polarization (TZV2P and TZVP for Ge and Te, respectively) and Goedecker-Teter-Hutter (GTH) pseudopotentials[73] were used for core electrons together with Perdew-Burke-Ernzerhof (PBE) exchange correlation functional[74]. All calculations were converged using a $10^{-7}$ Self-Consistent Field (SCF). Electronic potentials were calculated using a Poisson wavelet solver and periodic boundary conditions. Finally, the geometry optimization was performed using the Quickstep module, utilizing a Broyden-Fletcher-Goldfarb-Shannon (BFGS) optimizer. All geometry relaxations were run till RMS step and gradient convergence limit was reached. The electronic Density of States (DOS) was calculated by computing the phonon density of states (PDOS) of the 512-atom DFT relaxed structure and subsequently plotting a histogram of the energy eigenvalues. The Fermi energy was calculated to be 3.506 eV.

### Molecular dynamics simulation & phonon density of states calculations
In order to compute the vibrational spectrum of the amorphous GeTe structure, ab-initio molecular dynamics (AIMD) simulations were performed using the open-source CP2K program[72]. AIMD simulations were performed on an amorphous 216-atom unit cell with fixed lattice parameters and periodic boundary conditions. Simulations were performed using 10 fs time intervals at 700 K with a CSVR thermostat. Atomic positions were determined using the same parameters as described previously in the DFT calculations.

The phonon density of states was calculated according to: Yazdani et al. by computing the power spectrum of the mass-weighted position correlation function $r_i(t)$[75], which we determine from AIMD trajectories as described previously. The phonon density of states is then given by the sum of the partial density of states of all atoms:

$$g_i(\omega) = m_i \omega^2 \left| \mathcal{F}\{r_i(t)\} \right|^2 \frac{\hbar\omega}{k_B T} \left(1 + \left(e^{\frac{\hbar\omega}{k_B T}} - 1\right)^{-1}\right), g(\omega) = \sum_i g_{i(\omega)}, \quad (1)$$

where $m_i$ is the mass of atom i, and $\mathcal{F}\{\ldots\}$ is a Fourier transform. The $\hbar\omega/K_B T(1 + (e^{\hbar\omega/K_B T} - 1)^{-1})$ term corrects for the fact that in the AIMD simulations, the thermal occupation of a mode with frequency is $K_B T/\hbar\omega$, rather than the occupation given by Bose-Einstein statistics.

## Data availability
Source data are provided with this paper.

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

## Acknowledgements

Authors thank N. Yazdani, J. Clarysse, X. Zhao, D. Boskovic, and R. Bernini for assistance during synchrotron measurement shifts and the Swiss Light Source for the provision of beamtime on the SuperXAS beamline. We thank M. Mücklich for technical assistance, and M. Müller and M. Wuttig from RWTH Aachen for providing sputtered amorphous GeTe samples. Electron microscopy measurements were performed at the Scientific Center for Optical and Electron Microscopy (ScopeM) of the Swiss Federal Institute of Technology. This work was funded by European Research Council (ERC) under the European Union's Horizon 2020 research and innovation programme, grant agreement No. 852751.

## Author contributions

Conceptualization: MY, Methodology: MY, SW, Investigation: SW, OY, MY, DK, FS, OVS, PMA, Visualization: MY, SW, Supervision: MY, Writing—original draft: SW, MY, Writing—review & editing: SW, PMA, OVS, VW, MY.

## Funding

## Competing interests

The authors declare no competing interests.
