## [Peer Review File · Nature Communications]

REVIEWER COMMENTS

Reviewer #1 (Remarks to the Author):

The manuscript by Wintersteller et al. describing the key features of GeTe nanostructured phase change memory material highlights the solid state chemistry features of these materials at nanoscale via experimental synthesis, in-situ XAS and theoretical calculation.

Key novel results:

- the authors compared bulk microstructures with GeTe nanoparticles showing the high temperature dynamics starting from the same disordered fcc type sublattice with Ge chains. the crystallization process pass through multiple steps including the ideal GeTe glass phase.
- the nanoscale effect on crystalization and aging of GeTe shows higher crystallization temperature
- those are novel and highly important features for PCM devices and development of novel memory materials.

Validity:

I think that that work presented here is very important for the phase changing memory materials and a step forward to understand them.

I present here the few minor concerns that I have regarding this study:

- it is not clear for the reader why the GeTe must have (if they must have) a ZnS shell. the authors should argument the choice of having the shell and the choice of the composition of the shell.
- how is the shell influencing the crystallization mechanism? how do the authors think that the interphase is between the core and shell is behaving during the high temperature crystallization.

did the authors consider HRTEM or RAMAN to clarify these aspects?

Significance

the manuscript brings the GeTe materials one step forward for implementation into memory devices. the study shows the potential of these materials at nanoscale for < 10 nm memory cells which was not explain before in details and with adequate measurements.

Data and methodology

All the data and analysis are accurate and well described.

based on my evaluation, I suggest the publication of this manuscript after minor corrections and the core/shell concept is better explained and/or characterized.

Reviewer #2 (Remarks to the Author):

Wintersteller et al. present a very interesting study on the crystallization process and structural dynamics in GeTe (nanoparticles and bulk, separately). Most of the discussion is based on synchrotron data, particularly on in-situ high-temperature X-ray absorption spectroscopy. The authors correlate the collected data to Ge and Te's coordination environments in GeTe, allowing them to propose structural models for the amorphous and crystalline phases. Moreover, the authors make a significant effort to support their scientific arguments with DFT calculations and models. Besides, the computational model is quite simple and scalable.

In conclusion, the data presented in this article is of high importance not only for phase change memories but also for how crystallization can change at the nanoscale. Thus, I recommend publishing this work in Nature Communications after some minor corrections/comments are addressed.

1- Although the article has a good storyline, the structure of the paper needs revision. Some of the headings used to guide the reader don't represent what is discussed in the section, entailing confusion. Just as one example, in the "Structural units in amorphous GeTe" section, there is an important discussion about the structural units of crystalline GeTe. In fact, the paragraph starts with a discussion of the crystalline phase data. Then in this section, the authors also discuss the crystallization path. Maybe before the sentence "Having deduced the structural units in amorphous and crystalline GeTe, we now proceed to", another heading would be necessary, such as "Crystallization paths" to better understand the structure of the text. Such types of situations occur later on as well.

2- Figure 1 is engaging, but it contains lots of information, and the way it is organized makes it challenging to grasp the message. I would recommend that the authors split the figure in 2 and try to clarify concepts.

3- In Figure 1F, the Ge-Te₁ and Ge-Te₂ distances are written to be 2.86 and 3.15 Å. However, in the discussion (line 131), the expressed distances are 2.86 and 3.14 Å.

4- I recommend not using the words "slight" and "slightly" to discuss coordination. Ex. "where every Ge atom has a total coordination of 4 (Supplementary Tables 2 and 3), slightly lower than the theoretical coordination of 6". Implying that a change from the expected 6 to 4 coordination number (whether in a crystalline or amorphous system) is a very small deviation is not right (whatever the source of the deviation might be). There are many differences and implications between 4 and 6 coordination environments in the crystal field theory to use this kind of adverb/adjective.

5- Why did the authors use the sputtering technique to prepare microparticles? Is it due to the similarities with the industrial preparation of PCM? By checking previous literature (<https://pubs.acs.org/doi/10.1021/acs.chemmater.8b02702>), it can be observed that the crystallization phenomena indeed take place at lower temperatures when the particle size increases while the effect becomes more spontaneous. However, this effect is only notorious for particles with sizes between 4.5 to 10 nm. Then, why use larger particles when the particles with sizes around 100 nm reported in the mentioned paper (without the use of amide) could afford the same effect observed in the bulk?

6- The need for a ZnS shell coating the GeTe nanoparticles needs to be justified. I assume this is due to the effect that can have surface oxidation without such a protective layer. This is important because the shell plays a crucial role in the dynamics of the crystallization process. I would also suggest the authors mention the shell thickness.

7- The presence of vacancies, or defects, might play a role in the cleaving of Ge-Ge bonds and the formation of Te-Ge-Te bonds during the crystallization process. The EDX quantification shows the microparticles have an excess of Ge with respect to Te compared to the excess of Te found in the nanoparticles. Can the authors elaborate on this? Or do they consider the differences irrelevant?

9- The authors showed gradual changes towards a crystalline phase upon heating in nanoparticles, compared to the abrupt transition observed in microparticles. In that case, why would one be interested in nanoparticles for PCM? Is it because of the suppressed aging achievable in the nanoscale?

Reviewer #3 (Remarks to the Author):

This paper reports on an experimental and computational investigation of the structural properties and the crystallization kinetics of GeTe bulk microparticles and nanoparticles. GeTe is the parent compound of the GeSbTe alloys, an important family of phase-change materials (PCMs) used in electrical memories. PCMs also have potential applications in neuromorphic computing and nanophotonics. All of these devices exploit the ability of PCMs to switch rapidly between a crystalline and an amorphous state, as well as between intermediate partly crystalline states.

This paper proposes a new structural model of the amorphous state of GeTe, as well as a new crystallization mechanism. As far as I can say, experiments and simulations have been carried out carefully. I appreciate the fact that detailed information about the experimental and simulation protocols is provided. The results are interesting, although I am not fully convinced by some of their claims. The authors should address the following points before I can recommend publication in Nature Communications.

1. As the authors point out, in the literature there are many computational studies of GeTe based on molecular dynamics, where amorphous models were generated by simulated melt-quenching. These models share some similarities with the ones proposed here (presence of homopolar bonds and Ge-Ge chains), but also show significant differences, in that typically only a fraction of Ge atoms (20-30% using PBE functionals) has tetrahedral coordination, whereas the majority have defective octahedral coordination. The authors should carry out a more accurate comparison between their models and the ab-initio melt-quenched ones. It should also be stressed that it is well possible that the properties of amorphous GeTe depend on the preparation protocol: in fact, it has been shown (in a computational work) that as-deposited amorphous GST should have higher concentration of tetrahedral structures than melt-quenched models.

2. How does the energetics (energy per atom) of their amorphous models compare with crystalline GeTe? It should be possible to extract from the literature the typical energy difference per atom between melt-quenched amorphous and crystalline models to make a direct comparison between their amorphous models and melt-quenched ones.

3. There are several works in the literature (e.g. Refs. 31-33) claiming that the relaxed glass should not contain Ge-Ge bonds, nor tetrahedral structures, but only Peierls-distorted octahedral-like motifs. Obviously, the model of the ideal GeTe glass shown in Figure 4 looks quite different. The authors should comment on this. Again, a study of the energetics may shed light on this point. Also, the fact that their ideal amorphous model apparently has a quasi-crystalline Te sublattice should be discussed.

4. Did the authors try to crystallize their amorphous models by performing "brute-force" ab initio MD at high T (600-700 K)? In principle, these simulations would enable to test directly whether the proposed crystallization mechanism occurs (as long as the crystallization times are sufficiently short). 500-atom melt-quenched models typically crystallize on a time scale of hundreds of ps at these temperatures. Of course, I am aware of the fact that these simulations are very demanding computationally.

4. It would be interesting to perform additional analysis of the electronic and vibrational properties of the amorphous models. Did the authors compute the phonon density of states? Since tetrahedral structures should lead to high-frequency modes, it would be important to check whether their vibrational spectrum is compatible with Raman data available in the literature.

5. The authors draw some analogies between Ge chains and CH₂– unit in the alkane homologous organic series. Can this analogy provide some insight about bonding in amorphous GeTe? It has been claimed that such bonding is fundamentally different from the one in the crystalline state (covalent versus metavalent bonding) and this difference is responsible for the optical contrast between the two states. Do the authors agree with this statement? Since this topic is currently heavily debated, any new insight would be valuable.

6. I do not agree with the following statement in the abstract "In contrast, the amorphous structure is generally assumed to consist of a highly random ordering of atoms, quite distant to its crystalline counterpart." In the past 15 years, sustained efforts have been made to elucidate the similarities in local structure (octahedral motifs, 4-membered rings) between the two phases.

Reviewer #4 (Remarks to the Author):

The paper reports on an experimental analysis of the crystallization process of GeTe in micro and nanoparticles. This is the subject of several previous experimental and theoretical works since GeTe is a prototypical phase change material whose crystallization kinetics is exploited in memory devices.

Here the authors propose a picture for the crystallization process somehow against the common wisdom on this subject. In literature, the crystallization process of GeTe and GeSbTe is typically described in terms of reorientation of ABAB square rings (A=Ge/Sb, B=Te) that are the building blocks of the crystalline phase (beta-phase for GeTe) and which are present in amorphous phase as well.

On the contrary, the authors here suggest that crystallization is driven by a preordering of the fcc lattice of Te (not ABAB rings). This alternative view is interesting, but it is based on a single experimental information, namely the dependence of the Te-Te coordination in the second shell inferred from EXAFS. I am not an expert of EXAFS and therefore I cannot judge the sensitivity of the Te-edge EXAFS spectrum on the number of second neighbors. The authors should provide evidence on the solidity of this number as the fitting of EXAFS are not reported in the SI (on the other hand the Te-Te signal is absent in the XAS spectrum of a-GeTe in Figs. S2, S3).

Provided that this evaluation of the Te-Te CN in the second shell was reliable (to be clearly demonstrated by the authors) I'm not fully convinced by the scenario proposed here.

Still given its originality, I would not deny publication.

There are some other issues to be addressed

1) It is known that the crystallization temperature is strongly dependent on surface oxidation (see pag. 27 in Kooi and Wuttig, Adv. Mat. 1908302 (2020)). NPs are capped by ZnS, but what's about oxidation of the microparticles?

2) The authors suggest that annealing of NPs leads to a more octahedral like configuration of Ge atoms (weakening of short and long bonds alternation). I would naively conclude that this configuration is closer to the beta phase and that therefore it would not hinder nucleation as the author instead suggest.

3) The transformation in NPs mentioned in point 2 above looks like an accelerated drift. How this scenario could be reconciled with the claim of a lower drift in NPs? By the way, is there any experimental evidence that NPs (not nanowires or ultrathin films) feature a reduced drift?

4) The authors ascribe the higher stability of amorphous NPs to the ZnS capping which reduces the atomic mobility. In Ref. 13 the author presented a different view on the rise of the crystallization temperature with the decrease of NPs size. Do the authors think that the organic ligands (or the GeI₂ termination) in colloidal NPs would have a similar effect of the ZnS capping?

5) MD simulations of the crystallization of GeTe in literature clearly show nucleation and growth.

The kinetics of crystallization from MD and from experimental differential scanning calorimetry are well described by the Wilson-Frenkel model as an activated process controlled by the self-diffusion coefficient. Are these results consistent with the scenario proposed here?

6) On the DFT calculations: it is not surprising that the system amorphizes by starting from a configuration with Ge dimers in the octahedral site as this is a structure with very high energy (no room to host a dimer in the octahedral site). The authors compared the gofr of their amorphous model with experiments. A comparison with the amorphous models generated by quenching from the melt within the same theoretical framework, as available in literature, is also mandatory.

7) On the DFT calculations: I do not see a fcc-like lattice of Te in Fig. 3a, the Te network looks very disordered as it should. I wonder what the structure in the inset on Fig. 3e (BAD Te-Te-Te) refers to.

REVIEWER COMMENTS & ANSWERS

Reviewer #1 (Remarks to the Author):

The manuscript by Wintersteller et al. describing the key features of GeTe nanostructured phase change memory material highlights the solid state chemistry features of these materials at nanoscale via experimental synthesis, in-situ XAS and theoretical calculation.

Key novel results:

-the authors compared bulk microstructures with GeTe nanoparticles showing the high temperature dynamics starting from the same disordered fcc type sublattice with Ge chains. the crystallization process pass through multiple steps including the ideal GeTe glass phase.

-the nanoscale effect on crystalization and aging of GeTe shows higher crystallization temperature

-those are novel and highly important features for PCM devices and development of novel memory materials.

Response: We would like to thank the reviewer for taking the time to read and help improve our work. We appreciate that they acknowledge many of the novelties of this paper such as the nanoscale effects on crystallization and in turn the important impact it has on nanoscale PCM theory and devices. The minor concerns the reviewer makes are extremely valid and we have implemented the clarifications where needed, in particular explaining the significance and role of the nanoparticle ZnS shell.

Validity:

I think that that work presented here is very important for the phase changing memory materials and a step forward to understand them.

I present here the few minor concerns that I have regarding this study:

- it is not clear for the reader why the GeTe must have (if they must have) a ZnS shell. the authors should argument the choice of having the shell and the choice of the composition of the shell.

Response: The reviewer makes an important point, which was not made clear enough in the text, that the ZnS shell is not a key requirement for GeTe nanoparticles to exhibit PCM properties. We have therefore adapted the text (first paragraph of the results section) explaining that ZnS, often in combination with SiO₂, is an extremely popular, low-cost capping material able to withstand high temperatures present during phase transition whilst providing negligible optical losses in the visible and near-infrared spectrum. In our work, we are able to synthesize nanoparticles with a ZnS shell which has the additional benefit of encapsulating our nanoparticles preventing them from coalescing at elevated temperatures. This enables us to study the nanoscale crystallization mechanism at elevated temperatures for longer periods of time without losing nanoscale effects.

Choice of the composition of the shell: We want to minimize the NP shell thickness, so only the surface of the nanoparticle is covered with a ZnS monolayer, protecting the surface. This way our GeTe nanoparticle core remains intact with uniform composition. Our EDX measurements reveal a shell composition of only 7 at.% enabling us to estimate a shell thickness of less than a single monolayer, which is important to minimize the influence of the shell on XAS spectroscopy results.

- how is the shell influencing the crystallization mechanism? how do the authors think that the interphase is between the core and shell is behaving during the high temperature crystallization.

did the authors consider HRTEM or RAMAN to clarify these aspects?

Response: We thank the reviewer for making this point which we have now further clarified through the high temperature XRD (HTXRD) measurements (Supplementary Figure 8). We observe no structural differences during the crystallization between the ZnS encapsulated GeTe nanoparticles and the sputtered GeTe microparticles. From HTXRD we simply observe an increase in the crystallization temperature which has already been shown in the case for GeTe nanoparticles

with native oleic acid ligands (Yarema *et al.* Chem. Mater. 2018, 30, 17, 6134–6143). Instead, as explained in the Discussion Section of the main text, the ZnS shell plays an important role in the stability of the amorphous phase and strongly influences the strength of beta-relaxations (degree of aging) in GeTe. We expect the ZnS shell to increase confinement in the nanoparticle which in turn increases lattice stresses and reduces atomic mobility and hence delays aging leading to an improvement in memory data retention capabilities. This is supported by theoretical calculations on 3nm GeTe ultrathin films capped by $(\text{ZnS})_8(\text{SiO}_2)_2$. Due to the low ZnS concentration, our experimental observations and understanding from literature we believe that there is no need to perform additional HRTEM or Raman measurements.

Significance

the manuscript brings the GeTe materials one step forward for implementation into memory devices. the study shows the potential of these materials at nanoscale for < 10 nm memory cells which was not explain before in details and with adequate measurements.

Data and methodology

All the data and analysis are accurate and well described.

based on my evaluation, I suggest the publication of this manuscript after minor corrections and the core/shell concept is better explained and/or characterized.

Response: We are pleased that this reviewer recommends publication and sees the potential of nanoparticle phase change materials for memory applications.

Reviewer #2 (Remarks to the Author):

Wintersteller *et al.* present a very interesting study on the crystallization process and structural dynamics in GeTe (nanoparticles and bulk, separately). Most of the discussion is based on synchrotron data, particularly on in-situ high-temperature X-ray absorption spectroscopy. The authors correlate the collected data to Ge and Te's coordination environments in GeTe, allowing them to propose structural models for the amorphous and crystalline phases. Moreover, the authors make a significant effort to support their scientific arguments with DFT calculations and models. Besides, the computational model is quite simple and scalable.

In conclusion, the data presented in this article is of high importance not only for phase change memories but also for how crystallization can change at the nanoscale. Thus, I recommend publishing this work in Nature Communications after some minor corrections/comments are addressed.

Response: We are very pleased that this reviewer has enjoyed our work and recognized the high relevancy for the phase change field and great impact on nanoscale crystallization. Furthermore, we are encouraged that they appreciate the complementary theoretical calculations to support experimental synchrotron findings.

1- Although the article has a good storyline, the structure of the paper needs revision. Some of the headings used to guide the reader don't represent what is discussed in the section, entailing confusion. Just as one example, in the "Structural units in amorphous GeTe" section, there is an important discussion about the structural units of crystalline GeTe. In fact, the paragraph starts with a discussion of the crystalline phase data. Then in this section, the authors also discuss the crystallization path. Maybe before the sentence "Having deduced the structural units in amorphous and crystalline GeTe, we now proceed to", another heading would be necessary, such as "Crystallization paths" to better understand the structure of the text. Such types of situations occur later on as well.

Response: We thank that the reviewer for giving particular care to the structure and readability of the paper. We have taken their suggestion on splitting Figure 1 into two separate Figures which in

turn allows us to restructure the text, improving readability and understanding. The original Figure 1 and heading 'Structural units in amorphous GeTe' will be split into the following two parts: 1) 'Structural Units in Amorphous and Crystalline GeTe' containing only room temperature XAS characterizations and corresponding structural units, and 2) 'High temperature crystallization' which focuses on the XAS ramp measurements and findings. We trust that this has achieved the goal of improving understanding and readability of the paper.

2- Figure 1 is engaging, but it contains lots of information, and the way it is organized makes it challenging to grasp the message. I would recommend that the authors split the figure in 2 and try to clarify concepts.

Response: As in part mentioned above, we have taken the reviewers' advice to split Figure 1 and think that this has improved understanding of the individual Figures. We have also relabeled the subfigures.

3- In Figure 1F, the Ge-Te1 and Ge-Te2 distances are written to be 2.86 and 3.15 Å. However, in the discussion (line 131), the expressed distances are 2.86 and 3.14 Å.

Response: Thank you for catching this error. We have corrected the figure annotations to 3.14Å.

4- I recommend not using the words "slight" and "slightly" to discuss coordination. Ex. "where every Ge atom has a total coordination of 4 (Supplementary Tables 2 and 3), slightly lower than the theoretical coordination of 6". Implying that a change from the expected 6 to 4 coordination number (whether in a crystalline or amorphous system) is a very small deviation is not right (whatever the source of the deviation might be). There are many differences and implications between 4 and 6 coordination environments in the crystal field theory to use this kind of adverb/adjective.

Response: Thank you for pointing out this over-simplification. We have adapted the text to be more precise.

5- Why did the authors use the sputtering technique to prepare microparticles? Is it due to the similarities with the industrial preparation of PCM? By checking previous literature (<https://pubs.acs.org/doi/10.1021/acs.chemmater.8b02702>), it can be observed that the crystallization phenomena indeed take place at lower temperatures when the particle size increases while the effect becomes more spontaneous. However, this effect is only notorious for particles with sizes between 4.5 to 10 nm. Then, why use larger particles when the particles with sizes around 100 nm reported in the mentioned paper (without the use of amide) could afford the same effect observed in the bulk?

Response: The reason to use sputtered microparticles (and not nanoparticles with bulk like properties) is to compare the effects observed in nanoparticles to industrial standards which are commonly used in the field and better understood (in comparison to nanoparticles). Nanoparticle PCM is still a growing field within the community and therefore we wanted to reach a broader community with our paper and show the crystallization mechanism is not just valid in nanoparticles but also in commonly used sputtered microparticles. We hope this clarifies this point for the reviewer.

6- The need for a ZnS shell coating the GeTe nanoparticles needs to be justified. I assume this is due to the effect that can have surface oxidation without such a protective layer. This is important because the shell plays a crucial role in the dynamics of the crystallization process. I would also suggest the authors mention the shell thickness.

Response: As also commented by reviewer 1, in addition to preventing oxidation the purpose of the ZnS shell is to prevent nanoparticle coalescence at elevated temperatures for prolonged time periods (i.e., for annealing measurements). The ZnS shell does not change the crystallization mechanism, as observed clearly by HTXRD measurements (Supplementary Fig. 8), but instead strongly influences the degree of confinement in the nanoparticle improving the stability of the amorphous phase due to weaker beta-relaxation forces. This is now described in further detail in the discussion section.

Calculation of the shell thickness is a good point to mention and has been added in the SI (Supplementary Table 1) We can calculate the shell thickness from the atomic composition values determined from EDX, corresponding to a sub monatomic ZnS layer around GeTe nanoparticles. We design such small shell thickness to minimize the influence of the shell on XAS spectroscopy results.

7- The presence of vacancies, or defects, might play a role in the cleaving of Ge-Ge bonds and the formation of Te-Ge-Te bonds during the crystallization process. The EDX quantification shows the microparticles have an excess of Ge with respect to Te compared to the excess of Te found in the nanoparticles. Can the authors elaborate on this? Or do they consider the differences irrelevant?

Response: This is a valid concern since Ge:Te stoichiometry is known to affect the crystallization temperature of GeTe (Gwon et al. ACS Appl. Mater. Interfaces 2017, 9, 47, 41387–41396). However, since we observe the same crystallization temperature of 180 °C as stoichiometric 1:1 GeTe in both in-situ XRD and XAS ramp measurements, without the formation of Ge crystallites we assume that the small discrepancy in EDX composition (within the error bars) is irrelevant.

8- The authors showed gradual changes towards a crystalline phase upon heating in nanoparticles, compared to the abrupt transition observed in microparticles. In that case, why would one be interested in nanoparticles for PCM? Is it because of the suppressed aging achievable in the nanoscale?

Response: As correctly identified by the reviewer, nanoparticles can form different local coordination environments in the amorphous phase at elevated temperatures prior to crystallization (observed in XAS ramp measurements and highlighted in Figure 6). The reason to study and use nanoparticles for PCM is 3-fold: 1) Increasing demand for higher memory density requires smaller memory devices where nanoscale effects must be understood. 2) Nanoparticles offer greater tunability than bulk materials enabling memory specific application. We have already shown that tuning size can increase crystallization temperature enabling reliable memory application in harsh environments with elevated temperatures (e.g., automotive industry) and the composition of the nanoparticle surface can also influence crystallization temperature, crystallization kinetics and stability. Finally, as mentioned, suppressed aging is observed in nanoparticles which improves data retention properties. We thank the reviewer for highlighting this point and hence we have described these advantages in the text (Introduction and Discussion).

Reviewer #3 (Remarks to the Author):

This paper reports on an experimental and computational investigation of the structural properties and the crystallization kinetics of GeTe bulk microparticles and nanoparticles. GeTe is the parent compound of the GeSbTe alloys, an important family of phase-change materials (PCMs) used in electrical memories. PCMs also have potential applications in neuromorphic computing and nanophotonics. All of these devices exploit the ability of PCMs to switch rapidly between a crystalline and an amorphous state, as well as between intermediate partly crystalline states.

This paper proposes a new structural model of the amorphous state of GeTe, as well as a new crystallization mechanism. As far as I can say, experiments and simulations have been carried out carefully. I appreciate the fact that detailed information about the experimental and simulation protocols is provided. The results are interesting, although I am not fully convinced by some of their claims. The authors should address the following points before I can recommend publication in Nature Communications.

Response: We thank the reviewer for taking the time to read and evaluate our paper, providing very useful feedback to improve the quality of the work presented. We have performed additional theoretical calculations as requested (i.e., energy-per-atom calculation, electronic density of states and vibrational spectrum) and have also incorporated a complete, detailed comparisons to theoretical works in literature on melt-quench amorphous models. We believe that this has improved the clarity of the paper and provided a better comparison between our model and models discussed in literature.

1. As the authors point out, in the literature there are many computational studies of GeTe based on molecular dynamics, where amorphous models were generated by simulated melt-quenching. These models share some similarities with the ones proposed here (presence of homopolar bonds and Ge-Ge chains), but also show significant differences, in that typically only a fraction of Ge atoms (20-30% using PBE functionals) has tetrahedral coordination, whereas the majority have defective octahedral coordination. The authors should carry out a more accurate comparison between their models and the ab-initio melt-quenched ones. It should also be stressed that it is well possible that the properties of amorphous GeTe depend on the preparation protocol: in fact, it has been shown (in a computational work) that as-deposited amorphous GST should have higher concentration of tetrahedral structures than melt-quenched models.

Response: We fully agree with the reviewer's point that the structure and properties of amorphous GeTe depend on a preparation method. Therefore, we designed this study to compare two distinctly different GeTe materials, such as colloiddally synthesized nanoparticles and sputtered bulk thin films. We revealed clear similarities for GeTe across the scales, which point to the fact that e.g., the crystallization mechanism and the structure of ideal glass are universal and independent of the deposition method.

We are particularly grateful for the suggestion to compare our structure model with the ab-initio melt-quench structures, available in the literature. We included a detailed comparison of total and partial pair correlation functions, coordination environments for Ge and Te atoms, electronic and phonon density of states. With such verification of our structure model, the paper is now stronger. Specifically, we show that our model is extremely similar to literature AIMD models, with small differences, pointing to the fact that our model seems to depict a more amorphous GeTe structure (i.e., fewer 3-fold coordinated 'defective octahedra' for Ge and Te, energy per atom, DOS, etc.).

2. How does the energetics (energy per atom) of their amorphous models compare with crystalline GeTe? It should be possible to extract from the literature the typical energy difference per atom between melt-quenched amorphous and crystalline models to make a direct comparison between their amorphous models and melt-quenched ones.

Response: The reviewer makes a great point that studying the energetics of the amorphous system would add valuable insight to the potential landscape of the different amorphous models. In the supporting information of *A. E. Kheir et al. Physical Review Materials* 5.9 (2021): 095004, Table S1 highlights an energy difference per atom of 121.7 meV for the crystalline and amorphous phase (determined from AIMD melt-quench). In comparison, our DFT calculations reveal -164.6498 and -164.4845 eV/atom for the crystalline and amorphous structure respectively, corresponding to an energy difference of 165.3 meV/atom. This suggests that our initial GeTe structure is characteristic of a higher degree of amorphousness. We have added this comparison as a schematic in the SI.

3. There are several works in the literature (e.g. Refs. 31-33) claiming that the relaxed glass should not contain Ge-Ge bonds, nor tetrahedral structures, but only Peierls-distorted octahedral-like motifs. Obviously, the model of the ideal GeTe glass shown in Figure 4 looks quite different. The authors should comment on this. Again, a study of the energetics may shed light on this point. Also, the fact that their ideal amorphous model apparently has a quasi-crystalline Te sublattice should be discussed.

Response: The reviewer makes an interesting point. The structure of relaxed 'ideal glass' is a topic still under debate. What has been shown in literature, which we also observe with our XAS measurements, is that aging (i.e., slow annealing) reduces the concentration of Ge-Ge bonds and tetrahedral structures. Our results, however, suggest that it is not possible to realize the amorphous GeTe structure, in which all Ge atoms arrange in defected octahedra (i.e., no Ge-Ge bonds), because it will be a crystalline GeTe phase. Instead, the ideal GeTe glass structure still has a low concentration of Ge-Ge homopolar bonds. Our results show that once the concentration of these bonds becomes critically low, the system then undergoes the phase transition to the crystalline counterpart (crystallization mechanism in Figure 5).

As the reviewer mentions, our experimental XAS results point towards an ideal glass where the underlying Te sublattice is partially ordered. Specifically, for the GeTe bulk sample, we observe that a third of Te-Te bonds attain crystalline-like ordering (i.e., 4-out-of-12 Te-Te bonds become like in crystalline fcc-type Te sublattice). We then explain these experimental findings with our structure model. There are many ways, 4-out-of-12 Te-Te bonds may order, some of which we

draw in Supplementary Figure 22. For example, it could be an array of multiple supercritical nucleation centers (J. Phys. Chem. Lett. 2013, 4, 24, 4241–4246), such as unlinked Te octahedra (left panel of Supplementary Figure 22) or Ge₄Te₄ cuboids, which are geometrically correlated via Ge-Te-Ge bridges (right panel of Supplementary Figure 22). Main text schematics show it simplistically in 2 dimensional (100) plane and similar 2D arrangement is exemplified for another plan in the middle panel of Supplementary Figure 22. We extend the explanation of partially crystalline Te sublattice of the ideal GeTe glass model in the paper.

4. Did the authors try to crystallize their amorphous models by performing "brute-force" ab initio MD at high T (600-700 K)? In principle, these simulations would enable to test directly whether the proposed crystallization mechanism occurs (as long as the crystallization times are sufficiently short). 500-atom melt-quenched models typically crystallize on a time scale of hundreds of ps at these temperatures. Of course, I am aware of the fact that these simulations are very demanding computationally.

Response: We would like to thank the reviewer for this excellent idea, which we are currently planning for a follow-up publication. The reviewer is absolutely correct that such calculations are computationally demanding. Moreover, a good AIMD study should include a statistical analysis of multiple starting models, performed at different annealing temperatures in order to cover the different crystallization dynamics which occur at temperatures below T_g. As a result, it is a massive project, which we believe is beyond the scope of the current study.

4. It would be interesting to perform additional analysis of the electronic and vibrational properties of the amorphous models. Did the authors compute the phonon density of states? Since tetrahedral structures should lead to high-frequency modes, it would be important to check whether their vibrational spectrum is compatible with Raman data available in the literature.

Response: We have taken the suggestion of the reviewer to add this valuable comparison which has been added to the SI of the paper (Supplementary Figures 17-18). In general, we observe a similar electronic structure in comparison to melt-quench models. The main difference is that our amorphous model has a smaller band gap, which alludes again to the fact that our GeTe initial structure is more amorphous, hence contains more in-gap defects. Furthermore, comparing the vibrational spectra, we observe lower contributions from high frequency modes around 150 cm⁻¹, due to lower concentrations from Ge octahedral centers.

5. The authors draw some analogies between Ge chains and CH₂- unit in the alkane homologous organic series. Can this analogy provide some insight about bonding in amorphous GeTe? It has been claimed that such bonding is fundamentally different from the one in the crystalline state (covalent versus metavalent bonding) and this difference is responsible for the optical contrast between the two states. Do the authors agree with this statement? Since this topic is currently heavily debated, any new insight would be valuable.

Response: Our paper focuses primarily on amorphous GeTe structure, for which the covalent bonding is a consensus in the field. We support the existing literature on amorphous structure, but we, for the first time, reveal similarity to the organic molecules, which are characteristic of a mixture of polar C-H and non-polar C-C bonds. In analogy, amorphous GeTe structure can be presented as branched Ge-Ge chains, connected via Ge-Te-Ge bridges, demonstrating two intertwined networks of highly covalent bonds.

The crystalline structure of GeTe is confirmed as an array of distorted octahedra. Our experimental methods, however, cannot distinguish the nature of bonding in crystalline GeTe.

6. I do not agree with the following statement in the abstract "In contrast, the amorphous structure is generally assumed to consist of a highly random ordering of atoms, quite distant to its crystalline counterpart." In the past 15 years, sustained efforts have been made to elucidate the similarities in local structure (octahedral motifs, 4-membered rings) between the two phases.

Response: We fully agree with the reviewer, and we modify the abstract of the paper, where this sentence used to appear.

Reviewer #4 (Remarks to the Author):

The paper reports on an experimental analysis of the crystallization process of GeTe in micro and nanoparticles. This is the subject of several previous experimental and theoretical works since GeTe is a prototypical phase change material whose crystallization kinetics is exploited in memory devices.

Here the authors propose a picture for the crystallization process somehow against the common wisdom on this subject. In literature, the crystallization process of GeTe and GeSbTe is typically described in terms of reorientation of ABAB square rings (A=Ge/Sb, B=Te) that are the building blocks of the crystalline phase (beta-phase for GeTe) and which are present in amorphous phase as well.

On the contrary, the authors here suggest that crystallization is driven by a preordering of the fcc lattice of Te (not ABAB rings). This alternative view is interesting, but it is based on a single experimental information, namely the dependence of the Te-Te coordination in the second shell inferred from EXAFS. I am not an expert of EXAFS and therefore I cannot judge the sensitivity of the Te-edge EXAFS spectrum on the number of second neighbors. The authors should provide evidence on the solidity of this number as the fitting of EXAFS are not reported in the SI (on the other hand the Te-Te signal is absent in the XAS spectrum of a-GeTe in Figs. S2, S3).

Provided that this evaluation of the Te-Te CN in the second shell was reliable (to be clearly demonstrated by the authors) I'm not fully convinced by the scenario proposed here.

Still given its originality, I would not deny publication.

Response: We would like to thank the reviewer for their time and effort to help improve the quality of our paper. We are very pleased to hear that the reviewer acknowledges the novelty and originality of our work in how the structure and structural dynamics of amorphous GeTe is quantified.

We strongly believe our approach does not contradict the common wisdom in the field. Instead, we present a complimentary study with a new way to look at the structure of amorphous GeTe by analyzing the second coordination sphere, which in the case of GeTe, is a superposition of Te sublattice on the local ordering of Ge atoms. This viewpoint is complementary to the ABAB description of GeTe structure (i.e., via the first coordination sphere of Ge and Te). The reorientation of two ABAB rings may still indeed happen for example to form Ge₄Te₄ cuboids as nucleation centers, which our model predicts for relaxed GeTe glass structure (middle panel of Supplementary Figure 22). Notably, our model agrees particularly well with the previous literature in the field, e.g., the total and partial pair correlation functions, coordination environments for Ge and Te atoms, electronic and phonon density of states, etc. We acknowledge, this was not explicitly mentioned previously, and we have therefore added the clarifying comment to the main text with regards to the complementarity of our approach with respect to other analyses of the first coordination sphere of Ge and Te.

Regarding sensitivity of XAS to Te-Te coordination, this is a great point which we have now clarified. We have included a figure in the SI which specifically focuses on the XANES spectra of the Te-Te edge (Supplementary Figure 17). From this plot, it becomes clear that two peak features (M1 & M2, see Guda et al., npj Comput. Mater. 2021, 7, 203), which are characteristic of a fcc-type structure, become increasingly dominant as the samples are annealed. This is especially the case during early stages of the annealing where the overall structure is still far from the crystalline state and the only change is due to the ordering of the Te sublattice (Figure 5). We believe that this has demonstrated the element sensitive nature of our in-situ XAS technique.

We are not particularly sure what the reviewer means with regard to the lack of Te-Te signal in the EXAFS fitting in Figs. S2 and S3. Fig. S2 contains only the normalized XAS signals for the micro- and nanoparticle GeTe for the Ge and Te edge, where it is not possible to distinguish the contributions of specific bonds such as Te-Te. Figs. S3 and S4, on the other hand, contain the EXAFS fits for the Ge and Te edges shown in Fig. S2. For the Ge edge, we cannot measure any Te-Te interaction. For the Te-edge, we clearly observe the Te-Te shells (i.e., second coordination sphere) for the crystalline GeTe samples, while Te-Te signal is expectably missing for the amorphous GeTe samples (because of no ordering of the second coordination shell). Note that the

annealing XAS experiments (Figure 5c-d) show an evolution of the Te-Te shells as the amorphous structure preorders and then crystallizes.

The fitting of EXAFS spectra in Figs. S3 and S4 are reported in Tables S2-S5. We carried out the analogous fitting for all temperature range as the samples were heated with the constant ramp or annealed. These fitting results are presented graphically in Figures S6 and S13.

There are some other issues to be addressed

1) It is known that the crystallization temperature is strongly dependent on surface oxidation (see pag. 27 in Kooi and Wuttig, *Adv. Mat.* 1908302 (2020)). NPs are capped by ZnS, but what's about oxidation of the microparticles?

Response: The reviewer makes a good point with regards to surface oxidation which strongly affects crystallization temperature. In addition to sample preparation which was performed exclusively under air-free environments (N₂-filled glove box and sealed capillaries), we observe crystallization temperatures of exactly 180°C for the microparticles from both XRD and XAS ramp measurements, suggesting oxidation has not occurred. Furthermore, a key indication to sample oxidation in GeTe often leads to Ge and Te segregation, which would also be visible in in-situ XRD measurements. We believe that this is proof enough for the reviewer to reasonably dismiss the role of oxidation in the nanoparticles.

2) The authors suggest that annealing of NPs leads to a more octahedral like configuration of Ge atoms (weakening of short and long bonds alternation). I would naively conclude that this configuration is closer to the beta phase and that therefore it would not hinder nucleation as the author instead suggest.

Response: The reviewer makes a very valid remark, and we thank them for raising this concern. We indeed observe from the fitting of the Ge-Te₁ and Ge-Te₂ bonds of the nanoparticles during annealing that the bond distances become increasingly similar suggesting a structure of GeTe, which appears closer to the crystalline beta phase. It should be noticed that the annealing offset temperatures (in respect to crystallization points) are significantly larger for GeTe nanoparticles (T_x-50 °C) than in the case of GeTe bulk (T_x-10 °C). Therefore, we expect much faster crystallization for nanoparticles at similar offset temperatures, likely due to the increase in octahedral configurations of Ge in the beta phase, as pointed out by the reviewer.

Our arguments in the paper concern the opposite nanoscale effect, which delays nucleation in GeTe nanoparticles due to weaker beta-relaxation at the same absolute temperature. We modified the main text to make this point clearer.

3) The transformation in NPs mentioned in point 2 above looks like an accelerated drift. How this scenario could be reconciled with the claim of a lower drift in NPs? By the way, is there any experimental evidence that NPs (not nanowires or ultrathin films) feature a reduced drift?

Response: This is an excellent remark, and we thank the reviewer for bringing this closer to our attention. Our original discussion on this topic has followed the logic that our ZnS encapsulated NPs are confined in 3D in comparison to 1D confinement in ZnS capped thin films and hence should also feature lower atomic mobility and lower drift. Supporting this claim are ultrafast DSC measurements performed by *Chen et. al* (*Cryst. Growth Des.* 2018, 18, 2, 1041–1046). The authors have studied crystallization kinetics of ligand free GeTe nanoparticles and reveal a higher glass transition and hence lower fragility for GeTe NP in comparison to thin films. From these measurements the authors calculate crystal growth rates (Fig. 4) revealing '2–3 orders of magnitude lower crystal growth rate at temperatures approaching T_g in comparison with the GeTe thin films, indicating a higher stability of the amorphous phase of the GeTe NPs', supporting our claims.

In addition, it is important to note that our EXAFS findings do not portray the specific kinetics of crystallization but rather reveal the structural crystallization pathways in nanoparticles. We realize that this may have created slight confusion and hence we have adapted the discussion section to emphasize this point and relate our findings more directly with the aforementioned experimental observations hoping this clears any doubt.

4) The authors ascribe the higher stability of amorphous NPs to the ZnS capping which reduces the atomic mobility. In Ref. 13 the author presented a different view on the rise of the crystallization temperature with the decrease of NPs size. Do the authors think that the organic ligands (or the GeI2 termination) in colloidal NPs would have a similar effect of the ZnS capping?

Response: This is a great point by the reviewer and an ongoing area of research within our group. In principle, we believe that the surface termination of NPs has a profound effect on the stability of the amorphous phase and crystallization kinetics, depending on the degree of confinement the surface provides. From our current understanding we would logically argue that organic ligands would provide relatively weak confinement due to their high surface energy whereas GeI2 termination would lie somewhere in between the high confinement of ZnS shell and weak organic ligands. Unfortunately, since this is still ongoing research, we are not able to provide full detail on these effects and focus here on the comparison between confined NP and bulk (two extremes) in the scope of this paper.

The thermodynamic model, describing the rise of crystallization temperature for GeTe nanoparticles is complementary to the findings of this paper. Specifically, we determined that the change of entropy upon crystallization is smaller than the entropy of fusion, alluding to the existence of more stable atomic configurations in GeTe nanoparticles, such as reported here structure units with CN=5 (Figure 6).

5) MD simulations of the crystallization of GeTe in literature clearly show nucleation and growth.

The kinetics of crystallization from MD and from experimental differential scanning calorimetry are well described by the Wilson-Frenkel model as an activated process controlled by the self-diffusion coefficient. Are these results consistent with the scenario proposed here?

Response: We strongly agree with the literature that crystallization is driven by atomic diffusion enabling the formation of supercritical crystalline nuclei which in turn enables fast crystal growth. In general, XAS is not the ideal technique to study this as we measure average coordination environments and hence distinguishing between multiple small critical nuclei and one large nucleation seed can be difficult. There are however several observations which point to the same conclusion.

Small hints can be found by studying the evaluated MSRD of the EXAFS fits, in combination with the corresponding coordination numbers. In the case of GeTe microparticles, we can clearly observe the MSRD decrease during annealing measurements during the formation of the intermediate state. This is consistent with the Wilson-Frenkel model as this corresponds to an increase in global order, suggesting we have formed multiple small nucleation centers, from which the system then rapidly crystallizes. In the nanoparticle case, we expect this also to be the case but is much harder to see due to higher disorder in the system. Furthermore, due to the lower atomic mobility in the nanoparticles, crystallization does not occur as fast as in microparticle GeTe enabling us to observe different coordination environments where atoms are more 'stuck' in their local positions.

We acknowledge this is quite challenging to determine from XAS measurements alone but hope the insights provided help the reviewer understand our observations.

6) On the DFT calculations: it is not surprising that the system amorphizes by starting from a configuration with Ge dimers in the octahedral site as this is a structure with very high energy (no room to host a dimer in the octahedral site). The authors compared the gofr of their amorphous model with experiments. A comparison with the amorphous models generated by quenching from the melt within the same theoretical framework, as available in literature, is also mandatory.

Response: We thank the reviewer for highlighting this important point. We have included a comprehensive structural comparison of our amorphous model with melt-quench models found in literature in the SI. Specifically, from the comparison of gofr of our model with literature (REF: R. Mazzarello *et al. Phys. Rev. Lett.* 104, 085503 (2010)), we observe very similar distributions with the only differences being a much stronger Ge-Ge interaction and overall, slightly smaller bond distributions. Overall, however, we see that the gofr our model agrees very well with the literature.

7) On the DFT calculations: I do not see a fcc-like lattice of Te in Fig. 3a, the Te network looks very disordered as it should. I wonder what the structure in the inset on Fig. 3e (BAD Te-Te-Te) refers to.

Response: Figure 4a shows a 3D projection of the 512-atoms DFT-relaxed amorphous structure. However, when plotting exactly the same structure along the (100) viewing direction, the underlying fcc-type Te sublattice, although quite distorted, becomes apparent. In the inset of Figure 4e, we cut out a small portion of the 512-atom structure, but we agree that it would be more instrumental to show the whole Te sublattice instead. We therefore replace the inset of Figure 4e to include all Te-Te bonds of the 512-atom structure within the R_{cutoff} of 4.7 Å.

REVIEWERS' COMMENTS

Reviewer #1 (Remarks to the Author):

I am pleased with the corrections and the authors' answers. I think that the manuscript should be accepted as it is.

Reviewer #2 (Remarks to the Author):

The authors addressed all my comments. Paper is ready for publication.

Reviewer #3 (Remarks to the Author):

I think that the authors have addressed my concerns and the ones expressed by the other reviewers in a satisfactory way. I recommend the revised version for publication.

Reviewer #4 (Remarks to the Author):

The authors revised the manuscript to address the concerns raised by the referees. I am still not convinced about the scenario they propose. However,

for the reasons outlined in my first report I was not willing to deny publication at that time, nor I am going to do now.

I would just suggest the authors to take seriously the results they quoted in replying to the point 3 of the third referee. Their model of a-GeTe has a substantially higher energy than those generated by quenching from the melt (Fig. S16) which means that it cannot be identified with an 'ideal' glass, i.e. a more stable glass (also because of the large number of Ge-Ge bonds, in Fig. S14). Their model could be seen as an intermediate step toward crystallization which might require an activation energy to be reached from the more stable glassy configuration generated by quenching from the melt.

The very idea that a more reliable model of a-GeTe could be realized by the disordering procedure the authors propose seems misleading (as testified by its high energy). This does not exclude that the crystallization kinetics the authors propose with the intermediate locally Te-ordered model might be correct. Needless to say a convincing proof would come from AIMD crystallization of their amorphous model and of those generating by quenching from the melt to see if the same intermediate step appears as suggested by the third referee (point 4), but I understand that this is a computationally demanding task that the authors prefer not to address at this stage.

REVIEWER COMMENTS & ANSWERS

Reviewer #1 (Remarks to the Author):

I am pleased with the corrections and the authors' answers. I think that the manuscript should be accepted as it is.

Response: We are delighted to hear back and positive from the Reviewer. We would like to thank the Reviewer once again for their invaluable contribution to the peer-review process.

Reviewer #2 (Remarks to the Author):

The authors addressed all my comments. Paper is ready for publication.

Response: We are happy to have received a green light from the Reviewer. We appreciate Reviewer's time and effort dedicated to providing constructive feedback, which has undoubtedly strengthened the overall merit of our submission.

Reviewer #3 (Remarks to the Author):

I think that the authors have addressed my concerns and the ones expressed by the other reviewers in a satisfactory way. I recommend the revised version for publication.

Response: We express our sincere gratitude for the thoughtful review of our manuscript. Valuable comments and insightful suggestions have significantly contributed to the improvement of our work. We are delighted to receive positive feedback from the Reviewer.

Reviewer #4 (Remarks to the Author):

The authors revised the manuscript to address the concerns raised by the referees. I am still not convinced about the scenario they propose. However, for the reasons outlined in my first report I was not willing to deny publication at that time, nor I am going to do now.

I would just suggest the authors to take seriously the results they quoted in replying to the point 3 of the third referee. Their model of a-GeTe has a substantially higher energy than those generated by quenching from the melt (Fig. S16) which means that it cannot be identified with an 'ideal' glass, i.e. a more stable glass (also because of the large number of Ge-Ge bonds, in Fig. S14). Their model could be seen as an intermediate step toward crystallization which might require an activation energy to be reached from the more stable glassy configuration generated by quenching from the melt.

The very idea that a more reliable model of a-GeTe could be realized by the disordering procedure the authors propose seems misleading (as testified by its high energy). This does not exclude that the crystallization kinetics the authors propose with the intermediate locally Te-ordered model might be correct. Needless to say a convincing proof would come from AIMD crystallization of their amorphous model and of those generating by quenching from the melt to see if the same intermediate step appears as suggested by the third referee (point 4), but I understand that this is a computationally demanding task that the authors prefer not to address at this stage.

Response: We are particularly thankful to the Reviewer for their dedication to help improving our paper. We believe, however, that there has been a slight misunderstanding with respect to the simulated amorphous model and the structure of ideal glass this paper discusses. We would like to clarify that our simulated amorphous structure does not correspond in any form to our definition of an 'ideal glass'. The simulated amorphous structure is a representation of a highly disordered amorphous structure, which can be generated using computationally efficiently DFT relaxations of initial structure. On the other hand, the ideal glass structure is observed during annealing as partially stable intermediate state with partially ordered fcc-type Te sublattice.

Consequently, the Reviewer is completely correct that our simulated amorphous structure has substantially higher energy (i.e. is more disordered) than those generated by melt quench (e.g., Mazzarello et al. Phys Rev Lett 2010, 104, 085503) which is supported, as the Reviewer points out, by the high concentration of Ge-Ge bonds. The relevancy of our amorphous model is to demonstrate to the phase change memory community that reliable and intuitive representations of the amorphous structure can be obtained purely through DFT simulations, starting from an

ordered fcc-type Te sublattice with random orientations of homopolar Ge-Ge bonds within Te octahedra.

We do not believe (and have not mentioned) that our approach is more reliable than traditional melt-quench AIMD techniques. In our opinion these two approaches do not contradict but instead strengthen one another. We strongly acknowledge that in order to verify the structure of our observed intermediate state long time scale (& low temperature) AIMD simulations are required, which is an ongoing research topic in our group.

We once again would like to thank the reviewer for putting time and effort into improving the quality and understanding of the paper.